# OFFLINE POLICY LEARNING FOR NONPARAMETRIC CONTEXTUAL BANDITS UNDER RELAXED COVERAGE

## ABSTRACT

This paper is concerned with learning an optimal policy in a nonparametric contextual bandit from offline, and possibly adaptively collected data. Existing methods and analyses typically rely on i.i.d. offline data, and a uniform coverage condition on the behavior policy. In this work, similar to the single-policy concentrability coefficient, we propose a relaxed notion of coverage that measures how well the optimal action is covered by the behavior policy for the nonparametric bandits. Under this new notion, we develop a novel policy learning algorithm by combining the $k$-nearest neighbor method with the pessimism principle. The new algorithm has three notable properties. First and foremost, it achieves the minimax optimal suboptimality gap for any fixed coverage level (up to log factors). Second, this optimality is attained adaptively, without requiring prior knowledge of the coverage level of the offline data. Last but not least, it maintains these guarantees even with adaptively collected offline data.

## 1 INTRODUCTION

The contextual multi-armed bandit provides an elegant and powerful framework for modeling various sequential decision-making problems. In this setup, the learner engages with an environment in rounds: at each round, it observes contextual information, selects an action based on that context, and receives a reward corresponding to the chosen action. Notable advances in bandit algorithms over recent decades have led to successful applications in areas such as personalized medicine, online recommendation, and crowdsourcing (Kim et al., 2011; Li et al., 2010; Kittur et al., 2008).

The abundance of data from past deployments presents an opportunity to improve future decision-making via offline learning. This involves learning effective policies from batch datasets, a problem widely investigated in offline reinforcement learning (RL) (Lange et al., 2012; Levine et al., 2020; Chen & Jiang, 2019; Jin et al., 2021; Rashidinejad et al., 2021). Unlike online RL, which relies on active exploration, offline RL focuses on extracting insights from pre-collected data and has demonstrated effectiveness in high-stakes domains such as autonomous driving and healthcare (Bojarski, 2016; Yurtsever et al., 2020; Tang & Wiens, 2021).

Offline learning in the realm of contextual bandits has attracted growing attention in the past decade. Rashidinejad et al. (2021) studied the tabular case and Li et al. (2022); Zhu et al. (2023) subsequently considered offline learning in linear bandits. Another line of work (Swaminathan & Joachims, 2015; London & Sandler, 2019; Jin et al., 2022; Wang et al., 2024; Gabbianelli et al., 2024; Sakhi et al., 2024) advanced the understanding of parametric offline learning for contextual bandits by applying the importance-weighting method to learn a given policy class. While parametric offline learning has been extensively studied in the past literature, the problem of learning a policy from batch data under the nonparametric model remains underexplored.

The nonparametric contextual bandit is a classical personalized decision-making framework, where the expected reward for each action takes the form of a smooth function of the contexts (Yang & Zhu, 2002). For the online learning setting, the minimax rate of the cumulative regret has been established by (Perchet & Rigollet, 2013; Rigollet & Zeevi, 2010). Recent work Cai et al. (2024) initiated the study of nonparametric bandit learning with offline data. Their result displayed the ability of the offline dataset to improve the online learning regret. However, it imposed a uniform coverage condition on the offline dataset which says all actions need to be taken with sufficient

probability by the behavior policy in order for the offline data to be useful. Such uniform coverage condition may not hold in many practical scenarios.

In this paper, we study the problem of nonparametric offline policy learning under a relaxed coverage condition. Similar to the single-policy concentrability coefficient (Rashidinejad et al., 2021), we propose a relaxed notion of coverage that describes how well the optimal action is covered by the behavior policy for the nonparametric bandits. Since the quality of the offline data is often unknown in reality, it would be desirable to develop an algorithm that can learn an optimal policy from the batch data without knowing the coverage level. To that end, we introduce policy learning algorithms that bridge techniques from nonparametric statistics and the pessimism principle from offline RL (Jin et al., 2021; Rashidinejad et al., 2021). We summarize our theoretical results as follows.

## 1.1 MAIN CONTRIBUTIONS

First, we establish the fundamental limits of policy learning for nonparametric bandits under a relaxed coverage condition. Let $\beta$ be the smoothness parameter of the nonparametric reward functions (Assumption 2.1), and $\alpha$ be the margin parameter which measures the separation between the arms (Assumption 2.2). Denote by $C^\star$ the coefficient that reflects the coverage of the optimal action under the offline data (Definition 2.4), and denote by $d$ the dimension of the covariates. We show the minimax optimal suboptimality gap of policy learning for nonparametric bandits is of order $(N/C^\star)^{-\frac{\beta(1+\alpha)}{2\beta+d}}$, where $N$ is the size of the offline dataset. Intuitively, the coverage coefficient $C^\star$ controls the effective sample size of the offline data. When $C^\star$ becomes larger, the quality of the offline dataset degrades, and consequently the minimax rate of convergence decreases.

We introduce two nonparametric offline policy learning rules that nearly attain the optimal suboptimality gap. BIN-LCB (Algorithm 1) is based on splitting the continuous covariate space into smaller bins and applying the lower-confidence bound approach to each bin. By choosing the number of bins appropriately, we prove BIN-LCB achieves the optimal suboptimality up to log factors. A limitation of BIN-LCB is that the optimal binning parameter depends on the coverage coefficient $C^\star$, which is often unknown in practice.

To overcome this limitation, we propose KNN-LCB (Algorithm 2) that combines the $k$-nearest neighbor regression (Kpotufe, 2011; Chaudhuri & Dasgupta, 2014; Reeve et al., 2018) with the lower-confidence bound principle. The number of nearest neighbors considered at a covariate point is determined in a data-driven fashion, thereby allowing KNN-LCB to achieve the optimal suboptimality (up to log factors) without any knowledge of $C^\star$. Finally, in contrast to previous work that assume i.i.d. offline samples, our theoretical guarantees hold even when the batch dataset is generated by running adaptive algorithms.

## 1.2 RELATED WORK

**Nonparametric contextual bandits.** Since Woodroofe (1979) incorporated contextual information into the multi-armed bandits problem, there has been significant progress in the theory of contextual bandits. Auer (2002); Abbasi-Yadkori et al. (2011); Goldenshluger & Zeevi (2013); Bastani & Bayati (2020) adopted a parametric perspective and studied linear contextual bandits in both low and high dimensional settings. Meanwhile, modeling the reward function as a nonparametric function of the contexts was proposed by (Yang & Zhu, 2002). In the online learning setting, Rigollet & Zeevi (2010) developed a UCB-type algorithm that nearly achieves the optimal regret for nonparametric bandits and its results were further improved by (Perchet & Rigollet, 2013). Reeve et al. (2018) designed the $k$-Nearest Neighbor UCB algorithm that is able to utilize the low intrinsic dimensionality of the contexts. Cai et al. (2024) studied transfer learning for nonparametric bandits where the learner is given an offline dataset before starting to perform online learning. Additional insights in nonparametric bandits were developed in (Qian & Yang, 2016; Guan & Jiang, 2018; Hu et al., 2022; Suk & Kpotufe, 2021; Gur et al., 2022; Jiang & Ma, 2024).

**Offline policy learning.** Learning an optimal policy from batch data has received considerable attention in the past decade. Earlier works relied on the all-policy concentrability condition which requires the state-action pairs of all possible policies to be covered in the offline dataset (Munos & Szepesvári, 2008; Kallus, 2018; Chen & Jiang, 2019; Zhang et al., 2020; Xie & Jiang, 2021; Zhou et al., 2023). Nevertheless, such uniform coverage assumption can be violated in many practical

applications. A line of work relaxed this requirement by using the principle of pessimism to optimize conservatively on the batch data (Jin et al., 2021; Rashidinejad et al., 2021; Uehara & Sun, 2021; Xie et al., 2021a; Zanette et al., 2021; Shi et al., 2022; Zhan et al., 2022; Li et al., 2024a). For contextual bandits, Li et al. (2022); Zhu et al. (2023) studied pessimistic offline learning in linear bandits; Swaminathan & Joachims (2015); London & Sandler (2019); Jin et al. (2022); Wang et al. (2024); Gabbianelli et al. (2024); Sakhi et al. (2024) obtained policy learning guarantees via the importance weighting method when given a policy class. The prevalence of adaptive online learning algorithms has made it important to analyze situations where the offline dataset is adaptively collected (Zhan et al., 2024; Jin et al., 2022; Bibaut et al., 2021). Using both offline and online data to reduce sample complexity has been explored in hybrid RL (Xie et al., 2021b; Song et al., 2022; Wagenmaker & Pacchiano, 2023; Yang et al., 2023; Li et al., 2024c; Nakamoto et al., 2024). Finally, off-policy evaluation is a task related to policy learning but its main goal is to estimate the value function of a target policy based on offline data, and has been widely studied in recent literature (Thomas & Brunskill, 2016; Jiang & Li, 2016; Wang et al., 2017; Duan et al., 2020; Jiang & Huang, 2020; Ma et al., 2022; Li et al., 2024b; Lee & Ma, 2024).

## 2 PROBLEM SETUP

A $K$-armed nonparametric bandit instance is specified by a sequence of i.i.d. random vectors

$$(X_i, Y_i^{(1)}, \dots, Y_i^{(K)}), \quad 1 \leq i \leq N.$$

Here, each context $X_i$ is assumed to be from $\mathcal{X} := [0,1]^d$, and is sampled from a distribution $P_X$. We assume $P_X$ has a density (w.r.t. the Lebesgue measure) that is bounded below and above by some constants $\underline{c}, \bar{c} > 0$, respectively. In addition, $Y_i^{(a)}$ denotes the potential reward associated with choosing action $a$ at the $i$-th round. Denote by $\mathcal{I} = [K]$ the set of arms. Given any $a \in \mathcal{I}$ and $i \geq 1$, we assume that $Y_i^{(a)} \in [0,1]$ and

$$\mathbb{E}[Y_i^{(a)} \mid X_i] = f_a(X_i),$$

where $f_a$ is the unknown mean reward function for the arm $a$.

Suppose that we have access to an offline dataset $\mathcal{D}_{\text{off}} = \{(X_i, A_i, Y_i^{(A_i)})\}_{i=1}^N$ collected from a behavior policy $\mu = \{\mu_i\}_{i=1}^N$. More precisely, for $1 \leq i \leq N$, the action obeys $A_i \sim \mu_i(\cdot \mid X_i, \mathcal{F}_{i-1})$, where $\mu_i$ is the behavior policy at step $i$ that can depend on the history $\mathcal{F}_{i-1} = \{X_1, A_1, Y_1, \dots, X_{i-1}, A_{i-1}, Y_{i-1}\}$.

Define the optimal arm at $x$ to be

$$\pi^\star(x) \in \arg\max_{a \in \mathcal{I}} f_a(x),$$

with ties broken arbitrarily.

Our goal is to learn a policy $\pi : \mathcal{X} \to \mathcal{I}$ based on the batch dataset $\mathcal{D}_{\text{off}}$ that minimizes the expected suboptimality, which is defined as

$$\mathbb{E}\left[f_{\pi^\star(X)}(X) - f_{\pi(X)}(X)\right]. \tag{1}$$

### 2.1 ASSUMPTIONS

We adopt the following assumptions that are standard in nonparametric bandits (Rigollet & Zeevi, 2010; Perchet & Rigollet, 2013). The first assumption says the reward functions of different arms are Hölder smooth.

**Assumption 2.1** (Smoothness)**.** We assume that the reward function for each arm is $(\beta, L)$-smooth, that is, there exist $\beta \in (0,1]$ and $L > 0$ such that for any $a \in [K]$,

$$|f_a(x) - f_a(x')| \leq L\|x - x'\|_2^\beta, \quad \forall x, x' \in \mathcal{X}.$$

The second assumption depicts the separation of the reward functions of different arms. For any $x \in \mathcal{X}$, define the pointwise maximum of the reward functions at $x$ to be

$$f^\star(x) = f^{(1)}(x) = \max_{a \in [K]} f_a(x).$$

Besides, define the second pointwise maximum to be

$$f^{(2)}(x) = \max\{f_a(x) : a \in [K], f_a(x) < f^{(1)}(x)\},$$

if $f^{(1)}(x) \neq \min_{a \in [K]} f_a(x)$, and $f^{(2)}(x) = f^{(1)}(x)$ otherwise.

**Assumption 2.2** (Margin). We assume that the reward functions satisfy the margin condition with parameter $\alpha > 0$, that is there exist $C_\alpha > 0$ such that

$$P_X\left(0 < f^{(1)}(X) - f^{(2)}(X) \leq \delta\right) \leq C_\alpha \delta^\alpha, \quad \forall \delta \in [0, 1].$$

Assumption 2.2 originates from the margin condition in nonparametric classification (Mammen & Tsybakov, 1999; Tsybakov, 2004; Audibert & Tsybakov, 2007). It has been introduced to non-parametric contextual bandits by (Goldenshluger & Zeevi, 2009; Rigollet & Zeevi, 2010; Perchet & Rigollet, 2013). The complexity of the decision boundary is affected by the margin parameter. As the margin parameter $\alpha$ grows larger, the mean reward function of the optimal action becomes more well-separated from the other arms and identifying the optimal arm is less difficult.

*Remark* 2.3. When $\alpha\beta > 1$, the problem reduces to a static multi-armed bandit where one arm is always optimal, regardless of the context; see Proposition 2.1 in (Rigollet & Zeevi, 2010). In this degenerate case, the context becomes irrelevant, and the decision-making task loses its inherent complexity. Since our focus is on contextual bandits, we concentrate on the case $\alpha\beta \leq 1$ in this paper.

## 2.2 Relaxed coverage condition

To characterize the quality of the offline data, we define

$$\mu(\cdot \mid x) = \frac{1}{N} \sum_{i=1}^{N} \mu_i(\cdot \mid x), \tag{2}$$

which is the average of the behavior policies at $x$ over all time steps. We consider the following notion about the coverage of $\mu$.

**Definition 2.4.** We define $C^\star$ to be the positive constant that satisfies

$$\inf_{x \in \mathcal{X}} \mu(\pi^\star(x) \mid x) \geq \frac{1}{C^\star}. \tag{3}$$

Namely, $1/C^\star$ reflects the minimum probability that the optimal arm is taken over the covariate space. Compared to Cai et al. (2024), which requires the minimum probability that any arm is pulled to be lower bounded, our notion here is more relaxed because it only needs the optimal action to be covered. More precisely, one would have $\inf_{a \in [K], x \in \mathcal{X}} \mu(a \mid x) \leq \inf_{x \in \mathcal{X}} \mu(\pi^\star(x) \mid x)$ since the infimum is taken over all actions.

Definition 2.4 is related to the single-policy concentrability coefficient in offline tabular RL (Rashidinejad et al., 2021). Under the nonparametric bandit setting, however, the continuity of the covariate space together with the nonparametric reward function necessitates different techniques for the algorithm design and analysis of policy learning.

Throughout the paper, we assume the number of arms $K$ to be constant. We use $\mathcal{F}(\alpha, \beta, C^\star)$ to denote the family of $K$-armed bandit instances that satisfy the above conditions with parameters $\alpha$, $\beta$ and $C^\star$.

## 3 Minimax rates

The main challenge in learning an optimal policy from the batch dataset stems from the continuity of the covariate space. A common approach in nonparametric bandits literature is to decompose the state space into smaller bins and treat each bin as a multi-armed bandit problem without the covariate (Rigollet & Zeevi, 2010; Perchet & Rigollet, 2013). However, their algorithms are mainly designed for the online learning setup while in our case, the task is to learn a good policy from the offline data. The principle of pessimism has been a widely adopted method in offline reinforcement

---

**Algorithm 1** Binning with lower confidence bound

---

**Require:** Test point $x$, offline dataset $\mathcal{D}_{\text{off}}$, binning parameter $M$, confidence level $\delta$.
  1: Generate partition $\mathcal{L} \leftarrow \{B_j : j \in [M^d]\}$.
  2: Find $j \in [M^d]$ such that $x \in B_j$.
  3: Return $\hat{\pi}(x) = \arg\max_{a \in \mathcal{I}} \hat{f}_{a,j} - b_j(a)$.

---

learning (Jin et al., 2021; Rashidinejad et al., 2021). The key is to subtract an additional term from the empirical estimate of the reward value of an action to account for the uncertainty of the offline data. Motivated by these ideas, we propose Algorithm 1—Binning with lower confidence bound—to perform policy learning from the offline dataset for the nonparametric bandits.

To facilitate the presentation, we first introduce some notations. Let $\mathcal{L} = \{B_j : j \in [M^d]\}$ be a regular partition of $\mathcal{X}$ for some positive integer $M$, where

$$B_j = \{x \in \mathcal{X} : (v_l - 1)/M \le x_l < v_l/M, 1 \le l \le d\},$$

and $\boldsymbol{v} = (v_1, v_2, \ldots, v_d) \in [M]^d$. As a result, there are in total $M^d$ bins in $\mathcal{L}$. For any $j \in [M^d]$, let

$$N_j(a) = \sum_{i=1}^{N} \mathbf{1}\{X_i \in B_j, A_i = a\},$$

which is the number of times the covariate goes to bin $B_j$ and the action taken is $a$. Define

$$\hat{f}_{a,j} = \frac{1}{N_j(a)} \cdot \sum_{i=1}^{N} \mathbf{1}\{X_i \in B_j, A_i = a\} \cdot Y_i^a,$$

which is the empirical estimate of arm $a$'s reward in $B_j$. Define the uncertainty level of arm $a$ in $B_j$ to be

$$b_j(a) = \sqrt{\frac{2\log(1/\delta)}{N_j(a)}},$$

for some $\delta > 0$. For any $x \in \mathcal{X}$, Algorithm 1 first assigns it to the corresponding bin $B_j$. Then, it returns the action $a \in \mathcal{I}$ that maximizes $\hat{f}_{a,j} - b_j(a)$, which is the lower confidence bound of the reward value within that bin.

We are ready to present the theoretical guarantee of Algorithm 1.

**Theorem 3.1.** *Suppose $\alpha\beta \le 1$. Assume $N \ge (20\underline{c}^{-1}K\log(2N^5))^{(2\beta+d)/\beta}C^\star$. Algorithm 1 with inputs $M \asymp (N/C^\star)^{1/(2\beta+d)}$, $\delta = 1/N^5$ outputs a policy $\hat{\pi}$ that satisfies*

$$\sup_{\mathcal{F}(\alpha,\beta,C^\star)} \mathbb{E}\left[f_{\pi^\star(X)}(X) - f_{\hat{\pi}(X)}(X)\right] \le \widetilde{O}\left(\left(\frac{N}{C^\star}\right)^{-\frac{\beta(1+\alpha)}{2\beta+d}}\right).$$

See Section 5.1 for the proof.

The coverage coefficient $C^\star$ controls the effective sample size of the offline dataset. When $C^\star$ gets larger, which means the offline dataset has worse coverage on the optimal action, the expected suboptimality decays at a slower rate.

We complement the performance upper bound with the following minimax lower bound on the suboptimality of policy learning for nonparametric bandits.

**Theorem 3.2.** *Suppose $\alpha\beta \le 1$ and $C^\star \ge 2$. For any algorithm that takes in an offline dataset $\mathcal{D}_{\text{off}}$ and outputs a policy $\pi$, one has*

$$\sup_{\mathcal{F}(\alpha,\beta,C^\star)} \mathbb{E}\left[f_{\pi^\star(X)}(X) - f_{\pi(X)}(X)\right] \gtrsim \left(\frac{N}{C^\star}\right)^{-\frac{\beta(1+\alpha)}{2\beta+d}}.$$

See Appendix C for the proof.

Theorem 3.1 matches the lower bound in Theorem 3.2 up to log factors, and together they establish the minimax suboptimality gap of policy learning for nonparametric bandits under the relaxed coverage condition.

---

**Algorithm 2** KNN with lower confidence bound

---

**Require:** Test point $x$, offline dataset $\mathcal{D}_{\text{off}}$, confidence level $\delta$.
1: **for** each $a \in \mathcal{I}$ **do**
2:      $k(a) \leftarrow \arg\min_{k \in [N]} U_k^a(x)$.
3: **end for**
4: Return $\tilde{\pi}(x) = \arg\max_{a \in \mathcal{I}} \hat{f}_{k(a)}^a(x) - U_{k(a)}^a(x)$.

---

## 4   ADAPTIVITY TO THE COVERAGE COEFFICIENT

While Algorithm 1 nearly achieves the minimax optimal rate, it requires knowledge of the coverage coefficient $C^\star$ to determine the optimal binning parameter $M$. In practice, the quality of the offline dataset is often unknown, and the learner could face difficulties in choosing the appropriate value of $M$ without knowing $C^\star$. A natural attempt is to estimate the coverage coefficient. However, obtaining a faithful estimate of $C^\star$ is challenging because we do not know the optimal action at all. Instead of estimating $C^\star$ directly, we consider the following procedure that can optimally adapt to the quality of the batch dataset (Algorithm 2).

Our algorithm is inspired by the $k$-nearest neighbor UCB proposed in (Reeve et al., 2018). Their approach combines the $k$ nearest-neighbor method with the upper confidence bound to tackle the online regret minimization problem. In our case, however, the main challenge lies in adapting to the coverage of the offline dataset. On a high level, our procedure uses $k$-nearest neighbor regression to estimate the nonparametric reward functions, and applies the pessimism principle to select the action based on the batch data.

We start by introducing some notations. Given any $x \in \mathcal{X}$, let $\{\tau_q(x)\}_{q=1}^N$ be an enumeration of $[N]$ such that

$$\|x - X_{\tau_q(x)}\|_2 \leq \|x - X_{\tau_{q+1}(x)}\|_2,$$

for any $q \leq N - 1$. Denote by $\Gamma_k(x) = \{\tau_q : q \in [k]\}$ the set of indices of the $k$-nearest neighbors of $x$. Let

$$N_k^a(x) = \sum_{i \in \Gamma_k} \mathbf{1}\{A_i = a\},$$

which is the number of times arm $a$ is taken among the $k$-nearest neighbors of $x$. Let $r_k(x) = \|x - X_{\tau_k(x)}\|_2$. Define

$$U_k^a(x) = \sqrt{\frac{2 \log(1/\delta)}{N_k^a(x)}} + \log N \cdot r_k(x)^\beta,$$

which is the uncertainty value of arm $a$ with $k$ neighbors at $x$. Next, define

$$\hat{f}_k^a(x) = \frac{\sum_{i \in \Gamma_k} \mathbf{1}\{A_i = a\} Y_i^a}{N_k^a(x)},$$

which is the empirical estimate of arm $a$'s reward with the $k$-nearest neighbors of $x$ among the batch dataset. For any $x \in \mathcal{X}$, Algorithm 2 first determines a value $k(a)$ such that

$$k(a) = \arg\min_{k \in [N]} U_k^a(x), \tag{4}$$

for each $a \in \mathcal{I}$. Intuitively, $k(a)$ is the number of neighbors that can balance the bias and variance of the reward estimate at $x$ for arm $a$. Then, it selects the action $a \in \mathcal{I}$ that maximizes $\hat{f}_{k(a)}^a(x) - U_{k(a)}^a(x)$, which is the lower confidence bound of the reward estimate of arm $a$ at $x$. Now, we are ready to state the suboptimality guarantee of Algorithm 2.

**Theorem 4.1.** *Suppose $\alpha\beta \leq 1$. Assume $N \geq (20\underline{c}^{-1} K \log(2N^5))^{(2\beta+d)/\beta} C^\star$. Algorithm 2 with input $\delta = 1/N^5$ outputs a policy $\tilde{\pi}$ that satisfies*

$$\sup_{\mathcal{F}(\alpha,\beta,C^\star)} \mathbb{E}\left[f_{\pi^\star(X)}(X) - f_{\tilde{\pi}(X)}(X)\right] \leq \widetilde{O}\left((\frac{N}{C^\star})^{-\frac{\beta(1+\alpha)}{2\beta+d}}\right).$$

See Appendix B.1 for the proof.

Algorithm 2 attains the minimax optimal suboptimality up to log factors. In contrast to Algorithm 1, it does not need any knowledge of the coverage coefficient. As mentioned earlier, $C^\star$ dictates the effective sample size of the offline dataset. One can see from the proof of Theorem 1 that the optimal choice of the binning parameter $M$ balances the bias and variance in some sense based on the number of effective samples. In Algorithm 2, however, the burden of adapting to the effective sample size is left to the choice of $k$, the number of nearest neighbors to consider for reward estimation. Crucially, the definition of $k(a)$ in equation (4) allows for balancing the bias and variance of estimation in a data-driven fashion. Such choice of $k(a)$ in turn adapts to the local effective sample size at point $x$ for each arm $a \in \mathcal{I}$.

## 5 PROOF OF MAIN RESULTS

In this section, we present the analysis of Algorithm 1. While the full proof of our adaptive procedure (Algorithm 2) is postponed to Appendix B.1, the framework outlined here is instrumental for the later proof.

A key difficulty stems from the challenge of partial coverage in the presence of continuous covariates. With nonparametric reward functions, the optimal arm $\pi^\star(x) \in \arg\max_{a \in \mathcal{I}} f_a(x)$ can switch arbitrarily often, even within a tiny neighborhood, due to complex intersections among the reward curves $f_a(\cdot)$. Our key insight (Lemma 5.2) is the identification of at least one arm whose reward curve closely tracks the pointwise maximum $f^\star(\cdot)$ and, crucially, receives sufficient coverage under the behavior policy.

Moreover, a naïve application of existing pessimistic MAB bounds after partitioning the context space yields an excess risk guarantee that ignores the margin condition and is therefore loose. The novelty of our analysis lies in carefully controlling the error incurred on the regions where the best and second-best reward functions exhibit a sufficient gap. To do so, we further decompose the risk on those regions based on whether a near-optimal arm is selected or not, and derive high-probability bounds that can fully leverage the gap condition.

Motivated by (Perchet & Rigollet, 2013), we begin by partitioning the covariate space into different types of regions based on the separation of the reward functions. Define

$$\mathcal{J} = \{j \in [M^d] : \exists x_j \in B_j, f^{(1)}(x_j) - f^{(2)}(x_j) > cM^{-\beta}\}, \tag{5}$$

where $c > 0$ is to be specified. For its complement, we partition $\mathcal{J}^c$ into two smaller sets

$$\mathcal{J}_1^c = \{j \in \mathcal{J}^c : \exists x_j \in B_j, f^{(1)}(x_j) = f^{(2)}(x_j)\}, \tag{6}$$

and

$$\mathcal{J}_2^c = \{j \in \mathcal{J}^c : \forall x \in B_j, 0 < f^{(1)}(x) - f^{(2)}(x) \le cM^{-\beta}\}. \tag{7}$$

For any $j \in \mathcal{J}_1^c$, the following lemma shows that the regret incurred on $B_j$ can be controlled by the margin condition.

**Lemma 5.1.** *For any $j \in \mathcal{J}_1^c$ and policy $\pi : \mathcal{X} \to [K]$, one has*

$$\mathbb{E}\left[\sum_{j \in \mathcal{J}_1^c} \left(f^\star(X) - f_{\pi(X)}(X)\right) \mathbf{1}\{X \in B_j\}\right] \le C_\alpha \cdot c^{1+\alpha} \cdot M^{-\beta(1+\alpha)}.$$

See Section B.2.1 for the proof.

For any $j \in \mathcal{J} \cup \mathcal{J}_2^c$, let

$$\mathcal{I}_j^\star = \{a \in \mathcal{I} : \exists x \in B_j, f_a(x) = f^\star(x)\},$$

which is the set of near optimal arms in $B_j$. Our proof relies on the following observation that there always exists an arm in $\mathcal{I}_j^\star$ that is sufficiently covered by the offline dataset.

**Lemma 5.2.** *For any $j \in \mathcal{J} \cup \mathcal{J}_2^c$, there exists $a^\star \in \mathcal{I}_j^\star$ such that*

$$\mu(a^\star \mid B_j) = \frac{1}{P_X(B_j)} \int_{B_j} \mu(a^\star \mid x) \mathrm{d}P_X(x) \ge \frac{1}{KC^\star}.$$

See Section B.2.2 for the proof.

## 5.1 PROOF OF THEOREM 3.1

To start with, let $c_1 = 2\sqrt{\underline{c}^{-1}\log(2K\delta^{-1})K}$ and $c = c_1 + 2Ld^{\beta/2}$. For any $j \in \mathcal{J} \cup \mathcal{J}_2^c$, denote $a^\star \in \mathcal{I}_j^\star$ to be the near optimal arm with coverage in $B_j$ given by Lemma 5.2. For any $a \in \mathcal{I}$, define

$$f_{a,j} := \mathbb{E}[f_a(X) \mid X \in B_j] = \frac{1}{P_X(B_j)} \int_{B_j} f_a(x)\mathrm{d}P_X(x).$$

Deote $f_{\star,j} = f_{a^\star,j}$, and let $\hat{\pi}_j$ to be the output of Algorithm 1 on $B_j$. Define

$$\mathcal{A}_j = \{N_j(a^\star) \geq \frac{1}{2K}P_X(B_j) \cdot \frac{N}{C^\star}\},$$

and

$$\mathcal{E}_j = \{f_{\star,j} - f_{\hat{\pi}_j,j} \leq 2b_j(a^\star)\} \cap \mathcal{A}_j.$$

The next two lemmas state these good events happen with high probability.

**Lemma 5.3.** *For any $j \in \mathcal{J} \cup \mathcal{J}_2^c$, let $\hat{\pi}_j$ be the output of Algorithm 1 on $B_j$. With probability at least $1 - N^{-3}$,*

$$f_{\star,j} - f_{\hat{\pi}_j,j} \leq 2b_j(a^\star).$$

See Section B.2.3 for the proof.

**Lemma 5.4.** *Assume $\mu(a^\star \mid B_j) \geq 1/(C^\star K)$. One has*

$$\mathbb{P}(\mathcal{A}_j^c) \leq \frac{1}{N^5}.$$

See Section B.2.4 for the proof.

The excess risk can be decomposed as

$$\mathbb{E}\left[f^\star(X) - f_{\hat{\pi}(X)}(X)\right] = \sum_{j=1}^{M^d} \mathbb{E}\left[\left(f^\star(X) - f_{\hat{\pi}(X)}(X)\right)\mathbf{1}\{X \in B_j\}\right]$$

$$= \underbrace{\sum_{j \in \mathcal{J}} \mathbb{E}\left[\left(f^\star(X) - f_{\hat{\pi}(X)}(X)\right)\mathbf{1}\{X \in B_j\}\right]}_{U} + \underbrace{\sum_{j \in \mathcal{J}^c} \mathbb{E}\left[\left(f^\star(X) - f_{\hat{\pi}(X)}(X)\right)\mathbf{1}\{X \in B_j\}\right]}_{V}.$$

### 5.1.1 CONTROL OF TERM $V$

We further decompose

$$V = \underbrace{\sum_{j \in \mathcal{J}_1^c} \mathbb{E}\left[\left(f^\star(X) - f_{\hat{\pi}(X)}(X)\right)\mathbf{1}\{X \in B_j\}\right]}_{V_1} + \underbrace{\sum_{j \in \mathcal{J}_2^c} \mathbb{E}\left[\left(f^\star(X) - f_{\hat{\pi}(X)}(X)\right)\mathbf{1}\{X \in B_j\}\right]}_{V_2}.$$

For term $V_1$, Lemma 5.1 gives

$$V_1 \leq C_\alpha c^{1+\alpha} M^{-\beta(1+\alpha)}.$$

Next, we upper bound $V_2$. Fix any $j \in \mathcal{J}_2^c$. Let $a^\star \in \mathcal{I}_j^\star$ be the near optimal arm with coverage in $B_j$ given by Lemma 5.2, so that $\mu(a^\star \mid B_j) \geq 1/(KC^\star)$. Applying Lemma B.3 we get

$$\mathbb{E}[(f^\star(X) - f_{\hat{\pi}(X)}(X))\mathbf{1}\{X \in B_j\}] \leq 3\overline{c}cM^{-d-\beta}.$$

Besides, the margin condition implies

$$\sum_{j \in \mathcal{J}_2^c} \underline{c}M^{-d} \leq P_X(0 < f^{(1)}(x) - f^{(2)}(x) \leq cM^{-\beta}) \leq C_\alpha c^\alpha M^{-\beta\alpha}. \tag{8}$$

Therefore, $|\mathcal{J}_2^c| \leq \underline{c}^{-1}C_\alpha c^\alpha M^{d-\alpha\beta}$ and we reach

$$V_2 \leq \underline{c}^{-1}C_\alpha c^\alpha M^{d-\alpha\beta} \cdot (3\overline{c}cM^{-d-\beta}) = 3\underline{c}^{-1}\overline{c}C_\alpha c^{1+\alpha} M^{-\beta(1+\alpha)}.$$

Combining the bounds of $V_1$ and $V_2$ yields

$$V = V_1 + V_2 \leq (1 + 3\underline{c}^{-1}\overline{c})C_\alpha c^{1+\alpha} M^{-\beta(1+\alpha)}.$$

### 5.1.2 CONTROL OF TERM $U$

Fix any $j \in \mathcal{J}$. Let $\mathcal{I}_j^\star = \{a \in \mathcal{I} : f_a(x_j) = f^\star(x_j)\}$ where $x_j \in B_j$ satisfies $f^{(1)}(x_j) - f^{(2)}(x_j) > cM^{-\beta}$ by definition of $\mathcal{J}$. Let $a^\star \in \mathcal{I}_j^\star$ be the near optimal arm with coverage in $B_j$ given by Lemma 5.2, so that $\mu(a^\star \mid B_j) \geq 1/(KC^\star)$. Applying Lemma B.4 we obtain

$$\mathbb{E}[(f^\star(X) - f_{\hat{\pi}(X)}(X))\mathbf{1}\{X \in B_j\}]$$
$$\leq cM^{-\beta}\mathbb{P}(X \in B_j, 0 < f^{(1)}(X) - f^{(2)}(X) \leq cM^{-\beta}) + \mathbb{E}[\mathbf{1}\{X \in B_j, f_{\star,j} - f_{\hat{\pi}_j,j} \geq c_1 M^{-\beta}\}].$$
$$(9)$$

We further decompose the second term above into

$$\mathbb{E}[\mathbf{1}\{X \in B_j, f_{\star,j} - f_{\hat{\pi}_j,j} \geq c_1 M^{-\beta}\}]$$
$$= \mathbb{E}[\mathbf{1}\{X \in B_j, f_{\star,j} - f_{\hat{\pi}_j,j} \geq c_1 M^{-\beta}\}(\mathbf{1}\{\mathcal{A}_j\} + \mathbf{1}\{\mathcal{A}_j^c\})]$$
$$\leq \mathbb{E}[\mathbf{1}\{X \in B_j, f_{\star,j} - f_{\hat{\pi}_j,j} \geq c_1 M^{-\beta}\}\mathbf{1}\{\mathcal{A}_j\}] + \mathbb{P}(\mathcal{A}_j^c)$$
$$\leq \mathbb{E}[\mathbf{1}\{X \in B_j, f_{\star,j} - f_{\hat{\pi}_j,j} \geq 2b_j(a^\star)\}\mathbf{1}\{\mathcal{A}_j\}] + \mathbb{P}(\mathcal{A}_j^c)$$
$$\leq \frac{2}{N^3},$$
$$(10)$$

where the penultimate inequality uses the fact that $c_1 M^{-\beta} \geq 2b_j(a^\star)$ under $\mathcal{A}_j$, and the last inequality is due to Lemma 5.3 and Lemma 5.4. Combining relations (9) and (10), we reach

$$U = \sum_{j \in \mathcal{J}} \mathbb{E}\left[\left(f^\star(X) - f_{\hat{\pi}(X)}(X)\right)\mathbf{1}\{X \in B_j\}\right]$$
$$\leq \sum_{j \in \mathcal{J}} cM^{-\beta}\mathbb{P}(X \in B_j, 0 < f^{(1)}(X) - f^{(2)}(X) \leq cM^{-\beta}) + M^d \cdot \frac{2}{N^3}$$
$$\leq cM^{-\beta}\mathbb{P}(0 < f^{(1)}(X) - f^{(2)}(X) \leq cM^{-\beta}) + M^d \cdot \frac{2}{N^3}$$
$$\leq 2C_\alpha c^{1+\alpha} M^{-\beta(1+\alpha)},$$

where the last inequality uses the margin condition. Therefore,

$$\mathbb{E}\left[f^\star(X) - f_{\hat{\pi}(X)}(X)\right] = U + V$$
$$\leq 2C_\alpha c^{1+\alpha} M^{-\beta(1+\alpha)} + (1 + 3\underline{c}^{-1}\bar{c})C_\alpha c^{1+\alpha} M^{-\beta(1+\alpha)}$$
$$= (3 + 3\underline{c}^{-1}\bar{c})C_\alpha c^{1+\alpha} M^{-\beta(1+\alpha)}$$
$$= (3 + 3\underline{c}^{-1}\bar{c})C_\alpha c^{1+\alpha} \left(\frac{N}{C^\star}\right)^{-\frac{\beta(1+\alpha)}{2\beta+d}}.$$

## 6 DISCUSSION

In this paper, we establish policy learning guarantees for the nonparametric contextual bandits under a relaxed coverage condition which measures how well the optimal action is covered in the batch dataset. We design an adaptive procedure (Algorithm 2) that combines the $k$-nearest neighbors method with the pessimism principle to achieve the optimal suboptimality gap (up to log factors) without knowledge of the coverage coefficient.

Our work opens a few possible directions to pursue in the future. First, the current upper and lower bounds match up to log factors, and it would be interesting to remove the extra factors by sharpening the analysis. Besides, the smoothness parameter $\beta$ might be unknown in practice as well. Nevertheless, in the nonparametric bandit literature, it is widely acknowledged that adapting to the unknown smoothness parameter is generally impossible without additional assumptions on the reward functions (Locatelli & Carpentier, 2018; Gur et al., 2022). Understanding what conditions permit smoothness adaptation in offline policy learning is another direction to explore.

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

Figure 1: Suboptimality vs. $N$ ($C^\star = 4$).   Figure 2: Suboptimality vs. $C^\star$ ($N = 25000$).

# A EXPERIMENTS

We evaluate the proposed pessimistic nonparametric algorithms using a controlled synthetic contextual bandit environment with smooth, nonlinear reward functions and a logging policy with tunable coverage parameter $C^*$. Policies are learned from logged data of size $N$, then evaluated using true expected rewards on fresh test contexts. We report the suboptimality

$$\mathsf{Subopt}(\pi) = \mathbb{E}[f_{\pi^*(X)}(X) - f_{\pi(X)}(X)].$$

**Baselines.** We compare to two standard offline bandit baselines:

- KNN-GREEDY: A nonparametric estimator that chooses actions via $k$-nearest neighbor reward estimates, without uncertainty control.
- POEM (Swaminathan & Joachims, 2015): A self-normalized IPS method with a global variance penalty and a linear softmax policy class.

**Suboptimality vs. Sample Size.** Figure 1 shows suboptimality vs. $N$ at fixed coverage $C^\star = 4$. A few observations are in order.

- KNN-LCB achieves the lowest error across all sample sizes and decreases steadily with $N$.
- BIN-LCB is consistently second-best and displays a similar improvement trend.
- POEM performs worse than both pessimistic methods but is generally slightly better than KNN-GREEDY.
- KNN-GREEDY remains nearly flat as $N$ increases, reflecting its lack of pessimism.

Overall, the pessimistic nonparametric methods outperform both baselines, with gaps widening as the dataset grows.

**Suboptimality vs. Coverage.** Figure 2 shows suboptimality vs. $C^\star$ at fixed $N = 25,000$. Recall that as $C^\star$ increases, the coverage of the optimal action in the offline data decreases, leading to larger error.

- KNN-LCB achieves the lowest error for all coverage levels and degrades gracefully as $C^*$ increases.
- BIN-LCB exhibits similar behavior but maintains a consistent gap above KNN-LCB.
- POEM again outperforms KNN-GREEDY in general, but is still worse than the nonparametric methods.
- KNN-GREEDY is nearly insensitive to $C^*$ and remains dominated by both pessimistic methods.

These results demonstrate that the proposed algorithms adapt effectively to varying coverage, whereas POEM and KNN-GREEDY do not.

**Summary.** Across both experimental sweeps, KNN-LCB and BIN-LCB clearly outperform non-pessimistic and IPS-based baselines. POEM performs slightly better than KNN-GREEDY in several settings, but both are substantially worse than the pessimistic nonparametric methods, confirming the importance of local coverage-adaptive uncertainty quantification.

# B PROOF OF UPPER BOUNDS

## B.1 PROOF OF THEOREM 4.1

Define $M \asymp (N/C^\star)^{1/(2\beta+d)}$. Let $\{B_j\}_{j=1}^{M^d}$ be a regular partition of $\mathcal{X}$. Define $c_1 = 2(\sqrt{4\underline{c}^{-1}K\log\delta^{-1}} + \log N \cdot d^{\beta/2})$ and $c = c_1 + 2Ld^{\beta/2}$. Let $\mathcal{J}$, $\mathcal{J}_1^c$, $\mathcal{J}_2^c$ be defined as in equations (5), (6) and (7).

For any $j \in \mathcal{J} \cup \mathcal{J}_2^c$ and $x \in B_j$, let $k_j = \max\{q \in [N] : X_{\tau_q(x)} \in B_j\}$. Let $a^\star \in \mathcal{I}_j^\star$ be the near optimal arm with coverage in $B_j$ given by Lemma 5.2. Let $\tilde{\pi}(x)$ be the output of Algorithm 2. Define

$$\mathcal{A}_j(x) = \{N_{k_j}^{a^\star}(x) \geq \frac{1}{2K}P_X(B_j) \cdot \frac{N}{C^\star}\},$$

and

$$\mathcal{E}_j(x) = \{f_{a^\star}(x) - f_{\tilde{\pi}(x)}(x) \leq 2U_{k_j}^{a^\star}(x)\} \cap \mathcal{A}_j(x). \tag{11}$$

The next two lemmas state that these good events happen with high probability.

**Lemma B.1.** *Suppose $j \in \mathcal{J} \cup \mathcal{J}_2^c$ and $x \in B_j$. With probability at least $1 - N^{-3}$, one has*

$$f_{a^\star}(x) - f_{\tilde{\pi}(x)}(x) \leq 2U_{k_j}^{a^\star}(x).$$

See Section B.2.5 for the proof.

**Lemma B.2.** *Suppose $j \in \mathcal{J} \cup \mathcal{J}_2^c$ and $x \in B_j$. Assume $\mu(a^\star \mid B_j) \geq 1/(C^\star K)$. One has*

$$\mathbb{P}(\mathcal{A}_j^c(x)) \leq \frac{1}{N^5}.$$

See Section B.2.6 for the proof.

The excess risk can be decomposed as

$$\mathbb{E}\left[f^\star(X) - f_{\tilde{\pi}(X)}(X)\right] = \mathbb{E}\left[\sum_{j=1}^{M^d}\left(f^\star(X) - f_{\tilde{\pi}(X)}(X)\right)\mathbf{1}\{X \in B_j\}\right]$$

$$= \mathbb{E}\left[\underbrace{\sum_{j \in \mathcal{J}}\left(f^\star(X) - f_{\tilde{\pi}(X)}(X)\right)\mathbf{1}\{X \in B_j\}}_{U} + \underbrace{\sum_{j \in \mathcal{J}^c}\left(f^\star(X) - f_{\tilde{\pi}(X)}(X)\right)\mathbf{1}\{X \in B_j\}}_{V}\right].$$

### B.1.1 CONTROL OF TERM $V$

We further decompose

$$V = \underbrace{\sum_{j \in \mathcal{J}_1^c}\mathbb{E}\left[\left(f^\star(X) - f_{\tilde{\pi}(X)}(X)\right)\mathbf{1}\{X \in B_j\}\right]}_{V_1} + \underbrace{\sum_{j \in \mathcal{J}_2^c}\mathbb{E}\left[\left(f^\star(X) - f_{\tilde{\pi}(X)}(X)\right)\mathbf{1}\{X \in B_j\}\right]}_{V_2}.$$

For $V_1$, Lemma 5.1 gives

$$\mathbb{E}[V_1] \leq c^{1+\alpha}C_\alpha M^{-\beta(1+\alpha)}.$$

Next, we upper bound $V_2$. Fix any $j \in \mathcal{J}_2^c$. Define $\mathcal{I}_j^\star = \{a \in \mathcal{I} : \exists x \in B_j, f_a(x) = f^\star(x)\}$. Let $a^\star \in \mathcal{I}_j^\star$ be the near optimal arm with coverage in $B_j$ given by Lemma 5.2, so that $\mu(a^\star \mid B_j) \geq 1/(KC^\star)$. We have the following decomposition,

$$\mathbb{E}[(f^\star(X) - f_{\tilde{\pi}(X)}(X))\mathbf{1}\{X \in B_j\}] = \mathbb{E}[(f^\star(X) - f_{\tilde{\pi}(X)}(X))\mathbf{1}\{X \in B_j\}(\mathbf{1}\{\mathcal{E}_j(X)\} + \mathbf{1}\{\mathcal{E}_j^c(X)\})]$$

$$\leq \mathbb{E}[(f^\star(X) - f_{\tilde{\pi}(X)}(X))\mathbf{1}\{X \in B_j, \mathcal{E}_j(X)\}] + \mathbb{P}(X \in B_j, \mathcal{E}_j^c(X))$$

$$\leq \mathbb{E}[2U_{k_j}^{a^\star}(X)\mathbf{1}\{\mathcal{A}_j(X)\}]\bar{c}M^{-d} + \frac{2}{N^3}$$

$$\leq c_1 M^{-\beta} \cdot \bar{c}M^{-d} + \frac{2}{N^3} \leq 2\bar{c}c_1 M^{-d-\beta},$$

where the second inequality is due to the definition of $\mathcal{E}_j(X)$, and uses Lemma B.1 and Lemma B.2; the third inequality applies the definition of $\mathcal{A}_j(X)$. Reuse relation (8) we have $|\mathcal{J}_2^c| \leq \underline{c}^{-1}C_\alpha c^\alpha M^{d-\alpha\beta}$. Consequently,

$$\mathbb{E}[V_2] \leq \underline{c}^{-1}C_\alpha c^\alpha M^{d-\alpha\beta} \cdot \left(2\bar{c}c_1 M^{-d-\beta}\right) < 2\underline{c}^{-1}\bar{c}C_\alpha c^{1+\alpha}M^{-\beta(1+\alpha)}.$$

Combining the bounds of $V_1$ and $V_2$ yields

$$\mathbb{E}[V] = \mathbb{E}[V_1] + \mathbb{E}[V_2] \leq C_\alpha c^{1+\alpha}M^{-\beta(1+\alpha)} + 2\underline{c}^{-1}\bar{c}C_\alpha c^{1+\alpha}M^{-\beta(1+\alpha)} = (1 + 2\underline{c}^{-1}\bar{c})C_\alpha c^{1+\alpha}M^{-\beta(1+\alpha)}.$$

### B.1.2 CONTROL OF TERM $U$

Fix any $j \in \mathcal{J}$. Let $\mathcal{I}_j^\star = \{a \in \mathcal{I} : f_a(x_j) = f^\star(x_j)\}$ where $x_j \in B_j$ satisfies $f^{(1)}(x_j) - f^{(2)}(x_j) > cM^{-\beta}$ by definition of $\mathcal{J}$. Let $a^\star \in \mathcal{I}_j^\star$ be the near optimal arm with coverage in $B_j$ given by Lemma 5.2, so that $\mu(a^\star \mid B_j) \geq 1/(KC^\star)$. Applying Lemma B.4 we obtain

$$\mathbb{E}[(f^\star(X) - f_{\tilde{\pi}(X)}(X))\mathbf{1}\{X \in B_j\}]$$
$$\leq cM^{-\beta}\mathbb{P}(X \in B_j, 0 < f^{(1)}(X) - f^{(2)}(X) \leq cM^{-\beta}) + \mathbb{E}[\mathbf{1}\{X \in B_j, f_{a^\star}(X) - f_{\tilde{\pi}(X)}(X) \geq c_1 M^{-\beta}\}]. \tag{12}$$

We can further decompose the second term above into

$$\mathbb{E}[\mathbf{1}\{X \in B_j, f_{a^\star}(X) - f_{\tilde{\pi}(X)}(X) \geq c_1 M^{-\beta}\}]$$
$$= \mathbb{E}[\mathbf{1}\{X \in B_j, f_{a^\star}(X) - f_{\tilde{\pi}(X)}(X) \geq c_1 M^{-\beta}\}(\mathbf{1}\{\mathcal{A}_j(X)\} + \mathbf{1}\{\mathcal{A}_j^c(X)\})]$$
$$\leq \mathbb{E}[\mathbf{1}\{X \in B_j, f_{a^\star} - f_{\tilde{\pi}(X)} \geq c_1 M^{-\beta}\}\mathbf{1}\{\mathcal{A}_j(X)\}] + \mathbb{P}(\mathcal{A}_j^c(X))$$
$$\leq \mathbb{E}[\mathbf{1}\{X \in B_j, f_{a^\star} - f_{\tilde{\pi}(X)} \geq 2U_{k_j}^{a^\star}(X)\}\mathbf{1}\{\mathcal{A}_j(X)\}] + \mathbb{P}(\mathcal{A}_j^c(X))$$
$$\leq \frac{2}{N^3}, \tag{13}$$

where the penultimate inequality uses the fact that $c_1 M^{-\beta} \geq 2U_{k_j}^{a^\star}(x)$ under $\mathcal{A}_j(x)$; the last inequality is due to Lemma B.1 and Lemma B.2. Combining relations (12) and (13), we reach

$$\mathbb{E}[U] = \sum_{j \in \mathcal{J}} \mathbb{E}\left[(f^\star(X) - f_{\pi(X)}(X))\mathbf{1}\{X \in B_j\}\right]$$

$$\leq \sum_{j \in \mathcal{J}} \left(cM^{-\beta}\mathbb{P}(X \in B_j, 0 < f^{(1)}(X) - f^{(2)}(X) \leq cM^{-\beta}) + \frac{2}{N^3}\right)$$

$$\leq cM^{-\beta}\mathbb{P}(0 < f^{(1)}(X) - f^{(2)}(X) \leq cM^{-\beta}) + M^d \cdot \frac{2}{N^3}$$

$$\leq 2C_\alpha c^{1+\alpha}M^{-\beta(1+\alpha)},$$

where the last inequality is uses the margin condition. Therefore,

$$
\begin{aligned}
\mathbb{E}\left[f^{\star}(X) - f_{\tilde{\pi}(X)}(X)\right] &= \mathbb{E}[U] + \mathbb{E}[V] \\
&\leq 2C_{\alpha}c^{1+\alpha}M^{-\beta(1+\alpha)} + (1 + 2\underline{c}^{-1}\overline{c})C_{\alpha}c^{1+\alpha}M^{-\beta(1+\alpha)} \\
&= (3 + 2\underline{c}^{-1}\overline{c})C_{\alpha}c^{1+\alpha}M^{-\beta(1+\alpha)} \\
&= (3 + 3\underline{c}^{-1}\overline{c})C_{\alpha}c^{1+\alpha}\left(\frac{N}{C^{\star}}\right)^{-\frac{\beta(1+\alpha)}{2\beta+d}}.
\end{aligned}
$$

## B.2 PROOF OF HELPER LEMMAS

### B.2.1 PROOF OF LEMMA 5.1

For any $B_j \in \mathcal{J}_1^c$, there exists $x_j \in B_j$ such that $f^{(1)}(x_j) = f^{(2)}(x_j) = f_a(x_j)$ for all $a \in \mathcal{I}$. Consequently, the smoothness condition gives us $f^{(1)}(x) - f_a(x) \leq cM^{-\beta}$ for all $x \in B_j$. Since the set $\{x \in \mathcal{X} : f^{(1)}(x) = f^{(2)}(x)\}$ does not incur any error, we have

$$
\begin{aligned}
\sum_{j \in \mathcal{J}_1^c} \mathbb{E}\left[\left(f^{\star}(X) - f_{\pi(X)}(X)\right)\mathbf{1}\{X \in B_j\}\right] &\leq \sum_{j \in \mathcal{J}_1^c} cM^{-\beta}P_X\left(X \in B_j, 0 < f^{(1)}(X) - f^{(2)}(X) \leq cM^{-\beta}\right) \\
&\leq cM^{-\beta}P_X\left(0 < f^{(1)}(X) - f^{(2)}(X) \leq cM^{-\beta}\right) \\
&\leq C_{\alpha}c^{1+\alpha}M^{-\beta(1+\alpha)}.
\end{aligned}
$$

### B.2.2 PROOF OF LEMMA 5.2

By definition, for any $k \in \mathcal{I} \setminus \mathcal{I}_j^{\star}$, one has $f^{\star}(x) - f^k(x) > 0$ for all $x \in B_j$. Since

$$
\begin{aligned}
\sum_{a \in \mathcal{I}_j^{\star}} \mu(a \mid B_j) &= \frac{1}{P_X(B_j)} \sum_{a \in \mathcal{I}_j^{\star}} \int_{B_j} \mu(a \mid x)\mathrm{d}P_X(x) \\
&= \frac{1}{P_X(B_j)} \int_{B_j} \sum_{a \in \mathcal{I}_j^{\star}} \mu(a \mid x)\mathrm{d}P_X(x) \\
&\geq \frac{1}{P_X(B_j)} \int_{B_j} \frac{1}{C^{\star}}\mathrm{d}P_X(x) = \frac{1}{C^{\star}},
\end{aligned}
$$

where the inequality is due to $\sum_{a \in \mathcal{I}_j^{\star}} \mu(a \mid x) \geq 1/C^{\star}$, there exists $a^{\star} \in \mathcal{I}_j^{\star}$ such that

$$
\mu(a^{\star} \mid B_j) \geq \frac{1}{C^{\star}} \cdot \frac{1}{|\mathcal{I}_j^{\star}|} \geq \frac{1}{KC^{\star}}.
$$

### B.2.3 PROOF OF LEMMA 5.3

Denote $\mathcal{A}_j' = \{\hat{f}_{a,j} - b_j(a) \leq f_{a,j} \leq \hat{f}_{a,j} + b_j(a)$ for all $a \in \mathcal{I}\}$. By Lemma B.5, we have $\mathbb{P}(\mathcal{A}_j') \geq 1 - N^{-3}$. Since $\mathcal{A}_j'$ implies

$$
f_{\star,j} \leq \hat{f}_{\star,j} + b_j(a^{\star}) = \hat{f}_{\star,j} - b_j(a^{\star}) + 2b_j(a^{\star}) \leq \hat{f}_{\hat{\pi}_j,j} - b_j(\hat{\pi}_j) + 2b_j(a^{\star}) \leq f_{\hat{\pi}_j,j} + 2b_j(a^{\star}),
$$

we can conclude $\mathbb{P}(\{f_{\star,j} - f_{\hat{\pi}_j,j} \leq 2b_j(a^{\star})\}) \geq \mathbb{P}(\mathcal{A}_j') \geq 1 - N^{-3}$.

### B.2.4 PROOF OF LEMMA 5.4

For simplicity we drop the subscript on $j$ and write $B$ for $B_j$ throughout the proof. Recall $N_B(a^{\star}) = \sum_{i=1}^N \mathbf{1}\{X_i \in B, A_i = a^{\star}\}$. Denote $p_i = \mathbb{P}(X_i \in B, A_i = a^{\star})$. Define

$$
Z_i = \mathbf{1}\{X_i \in B, A_i = a^{\star}\} - \mathbb{E}[\mathbf{1}\{X_i \in B, A_i = a^{\star}\} \mid \mathcal{F}_{i-1}].
$$

One has $\mathbb{E}[\mathbf{1}\{X_i \in B, A_i = a^{\star}\} \mid \mathcal{F}_{i-1}] = p_i$, and it can be easily verified that $\{Z_i\}_{i=1}^N$ is a bounded martingale-difference sequence with $|Z_i| \leq 1$. Besides,

$$
\sum_{i=1}^N \mathbb{E}[Z_i^2 \mid \mathcal{F}_{i-1}] = \sum_{i=1}^N p_i(1 - p_i) \leq \sum_{i=1}^N p_i.
$$

By Freedman's inequality, we have

$$\mathbb{P}\left(|\sum_{i=1}^N Z_i| \geq \sqrt{2\left(\sum_{i=1}^N p_i\right)\log(\frac{2}{\delta})}\right) \leq \delta.$$

Therefore, with probability at least $1 - \delta$,

$$|N_B(a^\star) - P_X(B) \cdot N\mu(a^\star \mid B)| \leq \sqrt{3\log(\frac{2}{\delta})P_X(B) \cdot N\mu(a^\star \mid B)},$$

where we have used the relation $\mu(\cdot \mid x) = \frac{1}{N}\sum_{i=1}^N \mu_i(\cdot \mid x)$. Since $P_X(B) \cdot (N/C^\star) \geq 20K\log(2N^5)$ and $\delta = 1/N^5$, one has with probability at least $1 - 1/N^5$,

$$N_B(a^\star) \geq P_X(B) \cdot N\mu(a^\star \mid B) - \sqrt{3\log(\frac{2}{\delta})P_X(B) \cdot N\mu(a^\star \mid B)}$$

$$\geq \frac{1}{2}P_X(B) \cdot N\mu(a^\star \mid B) \geq \frac{1}{2K}P_X(B) \cdot \frac{N}{C^\star}.$$

### B.2.5 Proof of Lemma B.1

Denote $\mathcal{A}'_j = \{\hat{f}_{a,k} - U_k^a(x) \leq f_a(x) \leq \hat{f}_{a,k}(x) + U_k^a(x)$ for all $a \in \mathcal{I}, k \in [N-1]\}$. By Lemma B.6, we have $\mathbb{P}(\mathcal{A}'_j) \geq 1 - N^{-3}$. Under $\mathcal{A}'_j$, we have

$$f_{a^\star}(x) \leq \hat{f}_{a^\star,k(a^\star)}(x) + U_{k(a^\star)}^{a^\star}(x)$$

$$= \hat{f}_{a^\star,k(a^\star)}(x) - U_{k(a^\star)}^{a^\star}(x) + 2U_{k(a^\star)}^{a^\star}(x)$$

$$\overset{(i)}{\leq} \hat{f}_{\tilde{\pi}(x),k(\tilde{\pi}(x))}(x) - U_{k(\tilde{\pi}(x))}^{\tilde{\pi}(x)}(x) + 2U_{k(a^\star)}^{a^\star}(x)$$

$$\leq f_{\tilde{\pi}(x)}(x) + 2U_{k(a^\star)}^{a^\star}(x) \overset{(ii)}{\leq} f_{\tilde{\pi}(x)}(x) + 2U_{k_j}^{a^\star}(x),$$

where step (i) uses the definition of Algorithm 2, and step (ii) is due to $U_{k(a^\star)}^{a^\star}(x) \leq U_{k_j}^{a^\star}(x)$. Consequently,

$$\mathbb{P}(\{f_{a^\star}(x) - f_{\tilde{\pi}(x)}(x) \leq 2U_{k_j}^{a^\star}(x)\}) \geq \mathbb{P}(\mathcal{A}'_j) \geq 1 - N^{-3}.$$

### B.2.6 Proof of Lemma B.2

Recall $k_j = \max\{q \in [N] : X_{\tau_q(x)} \in B_j\}$. By definition, one has

$$N_{k_j}^{a^\star}(x) \geq \sum_{i=1}^N \mathbf{1}\{X_i \in B_j, A_i = a^\star\} = N_j(a^\star).$$

By Lemma 5.4,

$$\mathbb{P}(\mathcal{A}_j^c(x)) \leq \mathbb{P}(\mathcal{A}_j^c) \leq \frac{1}{N^5}.$$

### B.3 Auxiliary lemmas

**Lemma B.3.** *For any $j \in \mathcal{J}_2^c$, one has*

$$\mathbb{E}[(f^\star(X) - f_{\hat{\pi}(X)}(X))\mathbf{1}\{X \in B_j\}] \leq 3\overline{c}cM^{-d-\beta}.$$

*Proof.* We have the following decomposition,

$$\mathbb{E}[(f^\star(X) - f_{\hat{\pi}(X)}(X))\mathbf{1}\{X \in B_j\}] = \mathbb{E}[(f^\star(X) - f_{\hat{\pi}(X)}(X))\mathbf{1}\{X \in B_j\}(\mathbf{1}\{\mathcal{E}_j\} + \mathbf{1}\{\mathcal{E}_j^c\})]$$

$$\leq \mathbb{E}[(f^\star(X) - f_{\hat{\pi}(X)}(X))\mathbf{1}\{X \in B_j, \mathcal{E}_j\}] + \mathbb{P}(\mathcal{E}_j^c)$$

$$\leq \left(\mathbb{E}[(f_{\star,j} - f_{\hat{\pi}_j,j})\mathbf{1}\{\mathcal{E}_j\}] + cM^{-\beta}\right)\overline{c}M^{-d} + \mathbb{P}(\mathcal{E}_j^c),$$

where the last inequality uses $\mathbb{E}[f^\star(X) \mid X \in B_j] \le f_{\star,j} + cM^{-\beta}$ under the smoothness condition. Applying Lemma 5.3 and Lemma 5.4 , we reach

$$
\begin{aligned}
\mathbb{E}[(f^\star(X) - f_{\hat{\pi}(X)}(X))\mathbf{1}\{X \in B_j\}] &\le \left(\mathbb{E}[(f_{\star,j} - f_{\hat{\pi}_j,j})\mathbf{1}\{\mathcal{E}_j\}] + cM^{-\beta}\right)\overline{c}M^{-d} + \frac{2}{N^3} \\
&\le \left(\mathbb{E}[2b_j(a^\star)\mathbf{1}\{\mathcal{A}_j\}] + cM^{-\beta}\right)\overline{c}M^{-d} + \frac{2}{N^3} \\
&\le (c_1 + c)\,M^{-\beta} \cdot \overline{c}M^{-d} + \frac{2}{N^3} \le 3\overline{c}cM^{-d-\beta},
\end{aligned}
$$

where the second inequality is due to the definition of $\mathcal{E}_j$, and the third inequality uses the property of $\mathcal{A}_j$. $\square$

**Lemma B.4.** *Assume $c > 2Ld^{\beta/2}$. For any $j \in \mathcal{J}$ and any policy $\pi : \mathcal{X} \to [K]$, one has*

$$
\mathbb{E}[(f^\star(X) - f_{\pi(X)}(X))\mathbf{1}\{X \in B_j\}]
$$
$$
\le cM^{-\beta}\mathbb{P}(X \in B_j, 0 < f^{(1)}(X) - f^{(2)}(X) \le cM^{-\beta}) + \mathbb{E}[\mathbf{1}\{X \in B_j, f_{a^\star}(X) - f_{\pi(X)}(X) \ge (c - 2Ld^{\beta/2})M^{-\beta}\}].
$$

*Furthermore, if $\pi$ satisfies $\pi(x) = \pi_j \in [K]$ for all $x \in B_j$, one has*

$$
\mathbb{E}[(f^\star(X) - f_{\pi(X)}(X))\mathbf{1}\{X \in B_j\}]
$$
$$
\le cM^{-\beta}\mathbb{P}(X \in B_j, 0 < f^{(1)}(X) - f^{(2)}(X) \le cM^{-\beta}) + \mathbb{E}[\mathbf{1}\{X \in B_j, f_{\star,j} - f_{\pi_j,j} \ge (c - 2Ld^{\beta/2})M^{-\beta}\}].
$$

*Proof.* Recall $\mathcal{I}_j^\star = \{a \in \mathcal{I} : f_a(x_j) = f^\star(x_j)\}$. We have the following decomposition,

$$
\mathbb{E}[(f^\star(X) - f_{\pi(X)}(X))\mathbf{1}\{X \in B_j\}]
$$
$$
= \mathbb{E}[(f^\star(X) - f_{\pi(X)}(X))\mathbf{1}\{X \in B_j\}\mathbf{1}\{\pi(X) \in \mathcal{I}_j^\star\}] + \mathbb{E}[(f^\star(X) - f_{\pi(X)}(X))\mathbf{1}\{X \in B_j\}\mathbf{1}\{\pi(X) \in \mathcal{I} \setminus \mathcal{I}_j^\star\}]. \tag{14}
$$

For any $x \in B_j$ and $a \in \mathcal{I}_j^\star$, we have

$$
f^\star(x) - f_a(x) \le cM^{-\beta}\mathbf{1}\{0 < f^{(1)}(x) - f^{(2)}(x) \le cM^{-\beta}\}.
$$

So the first term in (14) can be bounded by

$$
\mathbb{E}[(f^\star(X) - f_{\pi(X)}(X))\mathbf{1}\{X \in B_j\}\mathbf{1}\{\pi(X) \in \mathcal{I}_j^\star\}] \le cM^{-\beta}\mathbb{P}(X \in B_j, 0 < f^{(1)}(X) - f^{(2)}(X) \le cM^{-\beta}).
$$

For any $a' \in \mathcal{I} \setminus \mathcal{I}_j^\star$ and $a \in \mathcal{I}_j^\star$, by definition $f_a(x_j) - f_{a'}(x_j) > cM^{-\beta}$. Consequently, the smoothness condition gives us

$$
f_a(x) - f_{a'}(x) \ge (c - 2Ld^{\beta/2})M^{-\beta} \tag{15}
$$

for all $x \in B_j$. Therefore, the second term in (14) can be bounded by

$$
\mathbb{E}[(f^\star(X) - f_{\pi(X)}(X))\mathbf{1}\{X \in B_j\}\mathbf{1}\{\pi(X) \in \mathcal{I} \setminus \mathcal{I}_j^\star\}] \le \mathbb{E}[\mathbf{1}\{X \in B_j, \pi(X) \in \mathcal{I} \setminus \mathcal{I}_j^\star\}]
$$
$$
\le \mathbb{E}[\mathbf{1}\{X \in B_j, f_{a^\star}(X) - f_{\pi(X)}(X) \ge (c - 2Ld^{\beta/2})M^{-\beta}\}].
$$

Since relation (15) implies $f_{a,j} - f_{a',j} \ge (c - 2Ld^{\beta/2})M^{-\beta}$, when $\pi(x) = \pi_j \in [K]$ for all $x \in B_j$, one has

$$
\mathbb{E}[(f^\star(X) - f_{\pi(X)}(X))\mathbf{1}\{X \in B_j\}\mathbf{1}\{\pi(X) \in \mathcal{I} \setminus \mathcal{I}_j^\star\}] \le \mathbb{E}[\mathbf{1}\{X \in B_j, \pi(X) \in \mathcal{I} \setminus \mathcal{I}_j^\star\}]
$$
$$
\le \mathbb{E}[\mathbf{1}\{X \in B_j, f_{\star,j} - f_{\pi_j,j} \ge (c - 2Ld^{\beta/2})M^{-\beta}\}].
$$

Combining all the above finishes the proof.

$\square$

**Lemma B.5.** *With probability at least $1 - N^{-3}$, one has*

$$
\hat{f}_{a,j} - b_j(a) \le f_{a,j} \le \hat{f}_{a,j} + b_j(a) \text{ for all } a \in \mathcal{I}.
$$

*Proof.* Fix any $a \in \mathcal{I}$. Recall

$$\hat{f}_{a,j} = \frac{1}{N_j(a)} \cdot \sum_{i=1}^{N} \mathbf{1}\{X_i \in B_j, A_i = a\} Y_i^a.$$

Denote $\epsilon_i = \mathbf{1}\{X_i \in B_j, A_i = a\}$, and $Z_i = \mathbf{1}\{X_i \in B_j\}(Y_i^a - f_{a,j})$. By Corollary 5 in (Reeve et al., 2018), one has

$$\mathbb{P}\left(|\hat{f}_{a,j} - f_{a,j}| > b_j(a)\right) = \mathbb{P}\left(|\sum_{i=1}^{N} \epsilon_i Z_i| > \sqrt{2\log(1/\delta)\sum_{i=1}^{N}\epsilon_i}\right) \leq e\left(\log(1/\delta)\log N\right)\delta.$$

Applying union bound we reach

$$\mathbb{P}((\mathcal{A}_j')^c) \leq K e\left(\log(1/\delta)\log N\right)\delta \leq \frac{1}{N^3}.$$

$\square$

**Lemma B.6.** *For any $x \in \mathcal{X}$, with probability at least $1 - N^{-3}$,*

$$\hat{f}_{a,k} - U_k^a(x) \leq f_a(x) \leq \hat{f}_{a,k}(x) + U_k^a(x) \text{ for all } a \in \mathcal{I}, k \in [N].$$

*Proof.* Fix $a \in \mathcal{I}$ and $k \in [N]$. Denote $\mathcal{G}_{a,k} = \{\hat{f}_{a,k} - U_k^a(x) \leq f_a(x) \leq \hat{f}_{a,k}(x) + U_k^a(x)\}$. On $\mathcal{G}_{a,k}^c$, one has $|\hat{f}_{a,k}(x) - f_a(x)| > U_k^a(x)$. Besides,

$$|\hat{f}_{a,k}(x) - f_a(x)| = |\frac{1}{N_k^a(x)} \sum_{s \in \Gamma_k} \left(\mathbf{1}\{A_s = a\}Y_s - f_a(x)\right)|$$

$$= |\frac{1}{N_k^a(x)} \sum_{s \in \Gamma_k} \left(\mathbf{1}\{A_s = a\}Y_s - f_a(X_s) + f_a(X_s) - f_a(x)\right)|$$

$$\leq |\frac{1}{N_k^a(x)} \sum_{s \in \Gamma_k} \left(\mathbf{1}\{A_s = a\}Y_s - f_a(X_s)\right)| + |\frac{1}{N_k^a(x)} \sum_{s \in \Gamma_k} \left(f_a(X_s) - f_a(x)\right)|$$

$$\leq |\frac{1}{N_k^a(x)} \sum_{s \in \Gamma_k} \left(\mathbf{1}\{A_s = a\}Y_s - f_a(X_s)\right)| + \log N \cdot r_k(x)^{\beta},$$

where the penultimate step uses triangle inequality, and the last inequality is due to

$$|f_a(X_s) - f_a(x)| \overset{(i)}{\leq} L\|x - X_s\|^{\beta} \overset{(ii)}{\leq} L \cdot r_k(x)^{\beta} \overset{(iii)}{\leq} \log N \cdot r_k(x)^{\beta}.$$

Here, step (i) is due to Assumption 2.1; step (ii) uses the definition of $r_k(x)$; step (iii) holds for $N$ sufficiently large. This leads to

$$\sqrt{\frac{2\log(1/\delta)}{N_k^a(x)}} + \log N \cdot r_k(x)^{\beta} = U_k^a(x) < |\frac{1}{N_k^a(x)} \sum_{s \in \Gamma_k} \left(\mathbf{1}\{A_s = a\}Y_s - f_a(X_s)\right)| + \log N \cdot r_k(x)^{\beta},$$

and we have

$$|\sum_{s \in \Gamma_k} \left(\mathbf{1}\{A_s = a\}Y_s - f_a(X_s)\right)| > \sqrt{2\log(1/\delta)N_k^a(x)}.$$

By Corollary 5 in (Reeve et al., 2018),

$$\mathbb{P}\left(|\sum_{s \in \Gamma_k} \left(\mathbf{1}\{A_s = a\}Y_s - f_a(X_s)\right)| > \sqrt{2\log(1/\delta)N_k^a(x)} \mid \{X_s\}_{s \in [N]}\right) \leq e\left(\log(1/\delta)\log N\right)\delta.$$

Therefore, using the law of total expectation we reach

$$\mathbb{P}(\mathcal{G}_{a,k}^c) \leq \mathbb{P}\left(|\sum_{s \in \Gamma_k} \left(\mathbf{1}\{A_s = a\}Y_s - f_a(X_s)\right)| > \sqrt{2\log(1/\delta)N_k^a(x)}\right) \leq e\left(\log(1/\delta)\log N\right)\delta.$$

Applying union bound to get

$$\mathbb{P}(\cup_{a \in [K], k \in [N]} \mathcal{G}_{a,k}^c) \leq e\left(\log(1/\delta)\log N\right) KN\delta \leq \frac{1}{N^3}.$$

$\square$

## C    PROOF OF THEOREM 3.2

**Step 1: introducing the family of problem instances.**    Take $P_X$ to be the uniform distribution on $\mathcal{X} = [0,1]^d$. Define $M = \lceil (N/C^\star)^{1/(2\beta+d)} \rceil$. Our construction of the reward instances is adapted from (Rigollet & Zeevi, 2010). Let $\mathcal{L} = \{B_j : j = 1, ..., M^d\}$ be a regular partition of $\mathcal{X}$ and let $q_j$ be the center of $B_j$. Denote $\Omega_m := \{\pm 1\}^m$ with $m := \lceil M^{d-\alpha\beta} \rceil$. For each $\omega \in \Omega_m$, define a function $f_\omega : [0,1]^d \mapsto \mathbb{R}$:

$$f_\omega(x) = \frac{1}{2} + \sum_{j=1}^m \omega_j \varphi_j(x), \tag{16}$$

where $\varphi_j(x) = C_\phi M^{-\beta}\phi(2M(x - q_j))\mathbf{1}\{x \in B_j\}$ with $\phi(x) = (1 - \|x\|_\infty)^\beta \mathbf{1}\{\|x\|_\infty \le 1\}$, and $C_\phi = \min(2^{-\beta}L, 1/4)$.

We take $K = 2$ and $\mathcal{I} = \{1, -1\}$. Let $\mu_i(1 \mid x) = 1/C^\star$ and $\mu_i(-1 \mid x) = 1 - 1/C^\star$ for all $x \in \mathcal{X}, 1 \le i \le N$. By equation (2), we have $\mu(1 \mid x) = 1/C^\star$ and $\mu(-1 \mid x) = 1 - 1/C^\star$ for any $x \in \mathcal{X}$. The family of problem instances of interest is

$$\mathcal{C} := \left\{ \left( \mu, f_1(x) = f_\omega(x), f_{-1}(x) = \frac{1}{2} \right) \mid \omega \in \Omega_m \right\}. \tag{17}$$

With slight abuse of notation, we also use $\mathcal{C}$ to denote $\{f_\omega : \omega \in \Omega_m\}$. It is straightforward to verify that $\mathcal{C} \subseteq \mathcal{F}(\alpha, \beta, C^\star)$.

**Step 2: reduction to the testing error.**    We can reduce lower bounding the suboptimality gap to the testing error due to to the following lemma.

**Lemma C.1** (Lemma 3.1 in (Rigollet & Zeevi, 2010)). *Under the margin condition, one has*

$$\mathbb{E}\left[ f^\star(X) - f_{\pi(X)}(X) \right] \ge \left( \frac{1}{D} \cdot \mathbb{E}\left[ 1\{\pi(X) \neq \pi^\star(X), f(X) \neq \frac{1}{2}\} \right] \right)^{\frac{\alpha+1}{\alpha}},$$

*for some constant $D > 0$.*

By Lemma C.1, we have

$$\sup_\mathcal{C} \mathbb{E}\left[ f^\star(X) - f_{\pi(X)}(X) \right] \ge \sup_\mathcal{C} (\frac{1}{D})^{\frac{\alpha+1}{\alpha}} \cdot \left( \mathbb{E}\left[ 1\{\pi(X) \neq \pi^\star(X), f(X) \neq \frac{1}{2}\} \right] \right)^{\frac{\alpha+1}{\alpha}}$$

$$= (\frac{1}{D})^{\frac{\alpha+1}{\alpha}} \cdot \left( \sup_\mathcal{C} \mathbb{E}\left[ 1\{\pi(X) \neq \pi^\star(X), f(X) \neq \frac{1}{2}\} \right] \right)^{\frac{\alpha+1}{\alpha}}.$$

Let $\mathbb{P}_\omega$ denotes the joint distribution of $\{(X_i, A_i, Y_i)\}_{i=1}^N$ under $\omega$, and let $\mathbb{E}_\omega$ be the corresponding expectation. The supreme term within the second parenthesis can be lowered bounded by the average,

$$\sup_\mathcal{C} \mathbb{E}\left[ 1\{\pi(X) \neq \pi^\star(X), f(X) \neq \frac{1}{2}\} \right] \ge \frac{1}{2^m} \sum_{\omega \in \Omega_m} \mathbb{E}_\omega\left[ 1\{\pi(X) \neq \pi^\star(X), f_\omega(X) \neq \frac{1}{2}\} \right]$$

$$= \frac{1}{2^m} \sum_{j=1}^m \sum_{\omega \in \Omega_m} \mathbb{E}_\omega\left[ 1\{\pi(X) \neq \omega_j, X \in B_j\} \right]$$

$$= \frac{1}{2^m} \sum_{j=1}^m \sum_{\omega_{[-j]} \in \Omega_{m-1}} \underbrace{\sum_{l \in \{\pm 1\}} \mathbb{E}_{\omega_{[-j]}^l}\left[ 1\{\pi(X) \neq l, X \in B_j\} \right]}_{W_{j,\omega_{[-j]}}}, \tag{18}$$

where $\omega_{[-j]}^l$ is the same as $\omega$ except for the $j$-th entry being $l$. Here we have used the fact that for $f_{\omega_{[-j]}^l}$, the optimal arm in the bin $B_j$ is $l$. We then relate $W_{j,\omega_{[-j]}}$ to a binary testing error. By Le

Cam's method,

$$W_{j,\omega_{[-j]}} = \frac{1}{M^d} \sum_{l\in\{\pm 1\}} \mathbb{P}_{\omega_{[-j]}^l}\left(\pi(X) \neq l \mid X \in B_j\right)$$

$$\geq \frac{1}{4M^d} \exp\left(-\text{KL}(\mathbb{P}_{\omega_{[-j]}^{-1}}, \mathbb{P}_{\omega_{[-j]}^1})\right)$$

$$\gtrsim \frac{1}{4M^d} \exp\left(-\frac{N}{C^\star} \cdot M^{-(2\beta+d)}\right),$$

where the last inequality uses Lemma C.2. Plugging the above back to equation (18) we obtain

$$\sup_{\mathcal{C}} \mathbb{E}\left[1\{\pi(X) \neq \pi^\star(X), f(X) \neq \tfrac{1}{2}\}\right] \geq \frac{1}{2^m} \sum_{j=1}^m \sum_{\omega_{[-j]}\in\Omega_{m-1}} W_{j,\omega_{[-j]}}$$

$$\gtrsim \frac{1}{2^m} \sum_{j=1}^m \sum_{\omega_{[-j]}\in\Omega_{m-1}} \frac{1}{4M^d} \exp\left(-\frac{N}{C^\star} \cdot M^{-(2\beta+d)}\right) \asymp M^{-\alpha\beta}.$$

Therefore, we can conclude the proof by

$$\sup_{\mathcal{C}} \mathbb{E}\left[f^\star(X) - f_{\pi(X)}(X)\right] \geq \left(\frac{1}{D}\right)^{\frac{\alpha+1}{\alpha}} \cdot \left(\sup_{\mathcal{C}} \mathbb{E}\left[1\{\pi(X) \neq \pi^\star(X), f(X) \neq \tfrac{1}{2}\}\right]\right)^{\frac{\alpha+1}{\alpha}}$$

$$\gtrsim (M^{-\alpha\beta})^{\frac{1+\alpha}{\alpha}} = M^{-\beta(1+\alpha)} \asymp \left(\frac{N}{C^\star}\right)^{-\frac{\beta(1+\alpha)}{2\beta+d}},$$

where we have used the definition of $M$.

**Lemma C.2.** *Fix $j \in [m]$. For any policy $\pi$, one has*

$$\text{KL}(\mathbb{P}_{\omega_{[-j]}^{-1}}, \mathbb{P}_{\omega_{[-j]}^1}) \lesssim \frac{N}{C^\star} \cdot M^{-(2\beta+d)}.$$

*Proof.* By the standard decomposition of the KL divergence and the Bernoulli reward structure,

$$\text{KL}(\mathbb{P}_{\omega_{[-j]}^{-1}}, \mathbb{P}_{\omega_{[-j]}^1}) \lesssim \sum_{i=1}^N \mathbb{E}_{\omega_{[-j]}^{-1}}\left[\left(f_{\omega_{[-j]}^{-1}}(X_i) - f_{\omega_{[-j]}^1}(X_i)\right)^2 \mathbf{1}\{A_i = 1\}\right]$$

$$\lesssim \sum_{i=1}^N M^{-2\beta} \mathbb{E}_{\omega_{[-j]}^{-1}}\left[\mathbf{1}\{A_i = 1, X_i \in B_j\}\right]$$

$$= \sum_{i=1}^N M^{-(2\beta+d)} \mathbb{P}_{\omega_{[-j]}^{-1}}\left(A_i = 1 \mid X_i \in B_j\right)$$

$$= \frac{N}{C^\star} \cdot M^{-(2\beta+d)}.$$

$\square$

