# OpenReview forum: "Offline Policy Learning for Nonparametric Contextual Bandits under Relaxed Coverage"
_ICLR.cc/2026/Conference — Submitted to ICLR 2026_

### Official Review · Reviewer_HC65 · 2025-10-17

**Soundness:** 4
**Presentation:** 3
**Contribution:** 3
**Rating:** 6
**Confidence:** 2

**Summary:**

This paper studies offline policy learning in nonparametric contextual bandits when the offline dataset may not satisfy the traditional uniform coverage condition. Specifically, it introduces a relaxed coverage assumption that focuses on how well the optimal action is covered by the behavior policy, extending the idea of the single-policy concentrability coefficient from offline RL. The work bridges the gap between nonparametric statistics and offline reinforcement learning, demonstrating that near-optimal policies can be learned from limited or adaptively collected data without strong coverage assumptions.

**Strengths:**

The introduction of the relaxed coverage notion is a key novelty. It generalizes prior assumptions and reflects more realistic conditions found in practice, e.g., healthcare or recommender systems where only near-optimal actions may have sufficient representation. The combination of nonparametric estimation with the pessimism principle is novel in this setting.

Theoretical contributions are rigorous. The authors establish tight minimax upper and lower bounds, which is significant because such tightness under partial coverage is nontrivial. The algorithms are derived systematically from the theoretical framework and are accompanied by detailed analyses. The proofs demonstrate a careful decomposition of the excess risk using margin and smoothness conditions, following classical nonparametric learning methods.

The paper is well-structured, starting with motivation and related work, followed by formal setup, algorithm descriptions, theoretical results, and detailed proofs.

The results make a meaningful theoretical advancement. They extend the landscape of offline learning from parametric to nonparametric models, achieving near-minimax rates without stringent assumptions.

**Weaknesses:**

The paper is entirely theoretical. While this is acceptable for a theory-focused venue, it would be valuable to include synthetic experiments demonstrating how the proposed algorithms perform with varying coverage levels. This would help illustrate the adaptivity of KNN-LCB in practice and bridge theory to application.

While the authors position the paper within offline policy learning, it could benefit from a stronger conceptual bridge to recent OPE results (e.g., minimax rates in OPE under limited coverage). This would help position the contribution more clearly relative to established theory in both RL and statistics.

The analysis assumes known smoothness and margin. The authors mention that adaptation to unknown smoothness is impossible without extra assumptions, but it would be helpful to provide more context, e.g., under what mild regularity assumptions adaptivity could be feasible.

**Questions:**

How can $C^\star$ be estimated or approximated from data in practice? Even if not required for the algorithm, understanding it empirically might provide diagnostic value.

---

> ### Author Response · Authors · 2025-11-20
>
> Thank you for dedicating time to read our paper and for your positive feedback.
>
> 1. On experiments:
>
> We have added an empirical validation section to the revised paper; see Appendix A (Experiments).
>
> 2. On conceptual bridge to recent OPE results:
>
> Thank you for the excellent suggestion. We will include a brief discussion of these connections in the revised version to clarify the relationship and further strengthen the positioning of our work.
>
> 3. On smoothness and margin:
>
> We would like to clarify that our algorithm only requires knowledge of the smoothness parameter. It does not take the margin parameter as input. The margin parameter appears solely in the analysis of the upper bound, where it is needed to convert local estimation error into global suboptimality and to establish the optimal rate.
>
> Regarding adaptation to the smoothness parameter, we note that the self-similarity condition is standard in the adaptive nonparametric bandit literature [1, 2]. Informally, it provides a global lower bound on the estimation bias of the reward functions. Under this condition, Lepski’s method [3] can be used to construct a reliable data-driven estimate of the smoothness level. This estimated smoothness can then be supplied to our algorithm in place of the true parameter. We expect that the price of such adaptation is limited to additional logarithmic factors in the upper bound, consistent with prior work.
>
> [1] Gur, Y., Momeni, A., and Wager, S. Smoothness-adaptive contextual bandits. (2022). Operations Research.
>
> [2] Cai, C., Cai, T. T., and Li, H. (2024). Transfer learning for contextual multi-armed bandits. The Annals of Statistics.
>
> [3] Lepski, O. V., Mammen, E., and Spokoiny, V. G. (1997). Optimal spatial adaptation to inhomogeneous smoothness: an approach based on kernel estimates with variable bandwidth selectors. The Annals of Statistics.
>
> 4. On estimating $C^\star$:
>
> We thank the reviewer for raising this point. As $C^\star = (\inf_x \mu(\pi^\star(x)\mid x))^{-1}$ depends on both the behavior policy and the unknown optimal policy, it cannot be estimated exactly without additional assumptions.Nevertheless, practical proxies can provide useful diagnostics, even though our adaptive KNN-LCB algorithm does not require $C^\star$.
>
> First, one can estimate the behavior policy $\mu(a\mid x)$ using flexible conditional density or classifier-based models, and evaluate the empirical quantity
> $\hat{C}_{\mathrm{local}} = 1 / \inf_x \hat{\mu}(\hat{\pi}(x)\mid x),$
> which measures how well the logged data support the actions chosen by the learned policy $\hat{\pi}$. This reflects the operational role of $C^\star$ in our analysis (i.e., governing the effective sample size along the policy’s decision boundary).
>
> Besides, in the binned version of our algorithm, one may estimate for each bin $B$, $p_{B,a} = \frac{n_{B,a}}{n_{B}}$,
> where $n_{B,a}$ is the number of logged samples in the bin choosing arm $a$. The minimum over bins of the arm chosen by $\hat{\pi}$ yields another conservative estimate of effective coverage.
>
> While none of these proxies estimate $C^\star$ itself, they provide interpretable, data-driven measures of effective coverage for the learned policy. We will add a brief discussion of such diagnostics in the revision.

---

> > ### Comment · Reviewer_HC65 · 2025-11-26
> >
> > Thank you very much for the useful clarifications and additional experiments, which address most of my concerns. I will maintain my positive assessment.

---

### Official Review · Reviewer_Qnj4 · 2025-10-24

**Soundness:** 3
**Presentation:** 3
**Contribution:** 2
**Rating:** 4
**Confidence:** 4

**Summary:**

This paper addresses the important problem of offline policy learning for nonparametric contextual bandits under a relaxed coverage condition. The relaxed coverage notion that only requires the optimal action to be covered, in contrast to the more stringent uniform coverage assumptions common in prior work. They introduce two algorithms: a binning-based method (BIN-LCB) and a more sophisticated k-nearest neighbor method (KNN-LCB) that incorporates pessimism. Theoretically, they establish a minimax lower bound and prove that both algorithms achieve a suboptimality gap that matches this lower bound up to logarithmic factors. A key strength of the KNN-LCB algorithm is its adaptivity—it achieves this optimal rate without prior knowledge of the coverage coefficient and its guarantees hold even for adaptively collected data.

The paper tackles a non-trivial and relevant problem, and the technical results appear sound and significant. The adaptive algorithm (KNN-LCB) is a particularly elegant solution to the practical challenge of unknown data quality.

**Strengths:**

- Minimax Optimality: The paper provides a complete minimax characterization of the problem. Theorem 3.2 establishes a fundamental lower bound, and Theorems 3.1 and 4.1 show that both proposed algorithms are minimax optimal up to log factors. This provides a clear picture of the problem's statistical limits.
- Adaptive Algorithm (KNN-LCB): The design of Algorithm 2 (KNN-LCB) is a major contribution. Its ability to adapt to the unknown coverage level in a data-driven manner is highly desirable for practical deployment. The mechanism of choosing
k(a) separately for each arm and test point to balance bias and variance is interesting
- Handling Adaptive Data: The theoretical guarantees are robust to adaptively collected offline data, which is a common scenario when the batch data comes from a previously deployed online learning algorithm. This significantly broadens the applicability of the results beyond the simpler i.i.d. setting.

**Weaknesses:**

- Lack of Empirical Validation: The most significant weakness of the paper is the complete absence of empirical results. Given that the KNN-LCB algorithm is a primary contribution, it is crucial to demonstrate its performance on synthetic or real-world datasets. How does it compare empirically to natural baselines (e.g., direct importance weighting, vanilla k-NN, or the non-adaptive BIN-LCB)? Does it indeed adapt to varying levels of coverage as theory predicts? Without empirical validation, it is difficult to assess the algorithm's practical utility and performance in finite-sample regimes.
- Assumption of Known Smoothness and Margin Parameters: The algorithms and their theoretical guarantees rely on prior knowledge of the smoothness parameter β and the margin parameter α. As the authors note in the discussion (Section 6), these are typically unknown in practice. While they cite literature suggesting that adaptation to unknown β is generally impossible, the paper would be significantly strengthened by a discussion or even a heuristic approach for setting these parameters, or an investigation of the consequences of misspecification.
- A naive implementation of Algorithm 2 would have a high computational cost. For a test point x, for each arm a, it must compute $U_k^a(x)$ for all k from 1 to N to find the minimizer k(a). This leads to a per-test-point complexity of $O(K * N)$, which is prohibitive for large datasets N. The paper should discuss computational considerations and potential approximations (e.g., using efficient nearest neighbor search, or searching over a subset of k values).

**Questions:**

- What happens if we do not assume the margin parameter? What is the role of margin parameter in this result and guarantees?

---

> ### Author Response · Authors · 2025-11-20
>
> Thank you for dedicating time to review our paper and for your detailed feedback.
>
> 1. “Lack of Empirical Validation”
>
> We have added an empirical validation section to the revised paper; see Appendix A (Experiments).
>
> 2. “Assumption of Known Smoothness and Margin Parameters”
>
> We would like to clarify that our algorithm only requires knowledge of the smoothness parameter. It does not take the margin parameter as input. The margin parameter appears solely in the analysis of the upper bound, where it is needed to convert local estimation error into global suboptimality and to establish the optimal rate.
>
> Regarding adaptation to the smoothness parameter, we note that the self-similarity condition is standard in the adaptive nonparametric bandit literature [1, 2]. Informally, it provides a global lower bound on the estimation bias of the reward functions. Under this condition, Lepski’s method [3] can be used to construct a reliable data-driven estimate of the smoothness level. This estimated smoothness can then be supplied to our algorithm in place of the true parameter. We expect that the price of such adaptation is limited to additional logarithmic factors in the upper bound, consistent with prior work.
>
> [1] Gur, Y., Momeni, A., and Wager, S. Smoothness-adaptive contextual bandits. (2022). Operations Research.
>
> [2] Cai, C., Cai, T. T., and Li, H. (2024). Transfer learning for contextual multi-armed bandits. The Annals of Statistics.
>
> [3] Lepski, O. V., Mammen, E., and Spokoiny, V. G. (1997). Optimal spatial adaptation to inhomogeneous smoothness: an approach based on kernel estimates with variable bandwidth selectors. The Annals of Statistics.
>
> 3. “A naive implementation of Algorithm 2 would have a high computational cost.”
>
> While the current presentation of KNN-LCB uses a full nearest-neighbor ordering with a naïve per-query cost linear in $N$, practical implementations are far more efficient. KNN-LCB only requires neighbors of a query point in increasing distance order, and this can be obtained efficiently by building a simple spatial index (e.g., a KD-tree or Ball tree) once on the dataset. Such indices typically reduce nearest-neighbor retrieval from linear time to something much closer to logarithmic or otherwise sublinear, leading to a per-query cost of $O(k_{\mathrm{max}}\log N )$, where $k_{\mathrm{max}} \ll N$ is a user-chosen neighborhood cutoff. The theory indicates that the optimal $k$ is roughly $N^{2\beta/(2\beta+d)}$, offering a reasonable guideline for setting $k_{\mathrm{max}}$. These choices reduce the effective computational load by several orders of magnitude without altering the algorithm’s behavior, ensuring that KNN-LCB is practical in real-world settings.
>
> 4. “What happens if we do not assume the margin parameter? What is the role of margin parameter in this result and guarantees?”
>
> The margin condition controls how often the best and second-best actions have nearly equal rewards: $P_{X}(0<f^{(1)}(X)-f^{(2)}(X)\leq\delta)\leq C_{\alpha}\delta^{\alpha}$.
>
> Intuitively, it quantifies the difficulty of distinguishing the optimal arm across contexts. When $\alpha$ is large, near-ties are rare and the decision boundary is well-separated; when $\alpha = 0$, ties are frequent and the learning problem becomes much harder.
>
> Our main results show that the achievable suboptimality rate is
> $\tilde{O}((N / C^\star)^{-\frac{\beta(1+\alpha)}{2\beta + d}})$,
> so $\alpha$ directly governs how quickly the optimal policy can be identified: larger $\alpha$ yields faster rates.
>
> If the margin condition is removed (i.e., $\alpha = 0$), our bound naturally reduces to the baseline rate $\tilde{O}( (N / C^\star)^{-\frac{\beta}{2\beta + d}})$, which matches the standard nonparametric regression rate and reflects the inherent ambiguity when optimal and suboptimal arms are frequently indistinguishable. In this regime, the learner can still estimate rewards accurately but cannot decisively select the optimal arm without additional separation.
>
> In short, the margin parameter is not required for consistency, and our algorithms remain valid for $\alpha = 0$. It shapes the difficulty of policy identification and determines the rate exponent in the minimax bound. The condition is standard in nonparametric contextual bandit analyses [4, 5] and provides a smooth transition between easy and hard regimes.
>
> [4] Audibert, J.-Y. and Tsybakov, A. B. Fast learning rates for plug-in classifiers. (2007). The Annals of Statistics.
>
> [5] Perchet, V. and Rigollet, P. The multi-armed bandit problem with covariates. (2013). The Annals of Statistics.

---

### Official Review · Reviewer_9v84 · 2025-10-25

**Soundness:** 3
**Presentation:** 3
**Contribution:** 3
**Rating:** 6
**Confidence:** 5

**Summary:**

This paper addresses the problem of offline policy learning for nonparametric contextual bandits. The authors consider a setting where the offline dataset may be collected adaptively and, crucially, relax the standard uniform coverage assumption to a much weaker single-policy concentrability condition that only requires the optimal action to be covered. Under this relaxed coverage and standard smoothness/margin conditions, the paper makes four key contributions: (1) It establishes the minimax optimal rate for the suboptimality gap, which is of order $$ \widetilde{O}\left(\left(N / C^{\star}\right)^{-\frac{\beta(1+\alpha)}{2 \beta+d}}\right) $$ ; (2) It proposes a binning-based algorithm (BIN-LCB) that achieves this rate but requires knowledge of the coverage coefficient $$C^{\star}$$ ; (3) Its main algorithmic contribution is a novel k-nearest neighbor-based algorithm (KNN-LCB) that adaptively achieves the minimax optimal rate without prior knowledge of $$C^{\star}$$ ; (4) All theoretical guarantees hold under adaptively collected data, significantly broadening their applicability.

**Strengths:**

The primary strength lies in formulating and solving the problem under a relaxed, single-policy concentrability condition, a major advancement over restrictive uniform coverage assumptions. The most notable innovation is the KNN-LCB algorithm, which is proven to be minimax-optimal without prior knowledge of the coverage coefficient $$C^{\star}$$ . Furthermore, the work establishes a complete minimax framework, providing matching upper and lower bounds, and all guarantees hold under adaptively collected data, enhancing the theory's rigor and practical relevance.

**Weaknesses:**

KNN-LCB requires an expensive procedure for each test point, involving searching over all data points for each action to determine the optimal number of neighbors, leading to a theoretical per-decision complexity of $$O(K \cdot N \cdot d)$$ . This is prohibitively expensive for large datasets.

**Questions:**

Given the computationally expensive nature of the KNN-LCB algorithm, which requires $O(K \cdot N \cdot d)$ operations per test point, do you see a pathway to significantly reduce this complexity to logarithmic or near-logarithmic scales, such as $O(K \cdot \log N \cdot d)$ ? For instance, could efficient data structures like KD-trees or approximate nearest neighbor techniques be integrated into your theoretical framework while preserving the minimax optimality guarantees?

---

> ### Author Response · Authors · 2025-11-20
>
> Thank you for dedicating time to read our paper and for your positive feedback.
>
> On the computational complexity of KNN-LCB:
>
> Although the theoretical presentation of Algorithm 2 describes a full nearest-neighbor ordering with a naïve cost linear in $N$ per query, practical implementations are far more efficient. The key point is that KNN-LCB only requires neighbors of a query point in increasing distance order, along with cumulative per-arm statistics over these neighbors. Thus, the main computational burden lies in nearest-neighbor retrieval. Building a simple spatial index (e.g., a KD-tree or Ball tree) once in $O(N d \log N)$ time reduces each query to $O(k_{\mathrm{max}}\log N )$, where $k_{\mathrm{max}} \ll N$ is a user-chosen cutoff on the neighborhood size. The theory indicates that the optimal $k$ is roughly $N^{2\beta/(2\beta+d)}$, offering a reasonable guideline for setting $k_{\mathrm{max}}$.
>
> Importantly, KNN-LCB does not require examining all $k \in {1,\dots,N}$. One can evaluate the uncertainty radii $U^a_k(x)$ only at a geometrically spaced set of candidate values (e.g., $1, 2, 4, 8, \dots, k_{\mathrm{max}}$), which captures the relevant scales while avoiding an exhaustive sweep. Cumulative per-arm counts and reward sums are updated incrementally as neighbors are added, keeping the overall per-query cost proportional to $k_{\mathrm{max}}$ with only a small dependence on the number of arms.
>
> In summary, with standard spatial indexing and a modest neighborhood cap, the effective per-query complexity of KNN-LCB is $O(k_{\mathrm{max}}\log N )$, not linear in $N$. These routine optimizations preserve both the structure and statistical guarantees of KNN-LCB, making the method computationally practical in real-world settings.

---

> > ### Comment · Reviewer_9v84 · 2025-11-24
> > **Thanks for your response**
> >
> > Thanks for your explanation on time complexity. Moreover, the following works may be related. Hope that authors can compare with the following works:
> >
> > Wanigasekara, Nirandika, and Christina Lee Yu. "Nonparametric contextual bandits in an unknown metric space." NeurIPS 2019.
> >
> > Puning Zhao et al. Zhao, Puning, et al. "Contextual bandits for unbounded context distributions." ICML 2025.

---

> > > ### Author Response · Authors · 2025-11-24
> > >
> > > Thank you for pointing out these works.
> > >
> > > Wanigasekara & Yu (2019) study nonparametric online contextual bandits in an unknown metric space, and design an adaptive partitioning algorithm whose regret depends on the local geometry of the reward functions. Their focus is on learning arm similarities and minimizing online regret when the metric over the context-arm space is unknown, whereas our work studies offline policy learning under a relaxed coverage condition, and characterizes the minimax suboptimality gap in terms of the coverage coefficient $C^\star$.
> > >
> > > Zhao et al. (2024) also consider nonparametric contextual bandits, but in the online setting with unbounded context distributions. They propose nearest-neighbor UCB algorithms with fixed and adaptive $k$, and derive minimax-optimal regret bounds that depend on the tail parameter of the context distribution. Our KNN-LCB procedure is technically related in that it combines nearest-neighbor estimation with confidence bounds, but the goals differ: (i) we operate in a purely offline setting, (ii) we optimize a pessimistic value estimate that accounts for coverage of the optimal action.
> > >
> > > We will add a discussion of these works to the Related Work section to clarify the distinctions.

---

### Official Review · Reviewer_7jZZ · 2025-10-27

**Soundness:** 3
**Presentation:** 2
**Contribution:** 3
**Rating:** 4
**Confidence:** 3

**Summary:**

This paper develops a nonparametric theoretical framework for offline policy learning in contextual bandits under relaxed coverage.
The setting considers a finite action set and continuous contexts, under smoothness and margin conditions on each action's reward function $f_a(x)$.
The authors introduce a coverage coefficient $1/C^\star$, quantifying the minimum probability that the optimal arm is taken over the covariate space, and derive the minimax lower bound on the suboptimality of policy learning:
\$$ \tilde O\left[\left(\tfrac{N}{C^\star}\right)^{-\frac{\beta(1+\alpha)}{2\beta+d}}\right],
\$$
They propose two nonparametric (pessimistic) algorithms: BIN-LCB, using binning-based local averages, and KNN-LCB, using adaptive knn regression, both combined with lower confidence bounds. Both (nearly) achieve the above rate. The work is entirely theoretical, extending pessimistic offline learning analyses to the nonparametric regime.

**Strengths:**

**S1.** Clean minimax characterization of policy learning performance under relaxed coverage and smooth reward assumptions.

**S2.** Rarely explored nonparametric perspective of offline policy learning in contextual bandits.

**S3.** Conceptually easy-to-understand algorithms (BIN-LCB and KNN-LCB) that enjoy good theoretical guarantees.

**Weaknesses:**

**W1. No empirical validation** is provided, even though offline contextual bandits are typically evaluated on practical benchmarks. Almost all papers in offline policy learning (even highly theoretical ones) include empirical results to illustrate performance.

**W2. Limited practicality of the proposed algorithms.** BIN-LCB has very high space complexity when $M$, $d$, or $K$ are large, and KNN-LCB incurs high inference time that scales with the dataset size $N$. As a result, the methods are practical only in low-dimensional, low-data regimes. Connecting to the point above, the absence of experiments prevents any demonstration of their strengths.

**W3. Insufficient discussion of the coverage condition.** The paper provides no clear interpretation of what happens when coverage is already sufficient (large $1/C^\star$). The lower-bound theorem assumes $C^\star$ is large enough: i.e., that the problem is sufficiently hard. This is understandable, so that we can derive the lower bound, but it should be discussed, especially whether the proposed algorithms become overly conservative in that case.

**W4. Unclear dependence on the number of actions.** The theoretical bounds do not make the scaling in $K$ explicit, even though this dependence is important in offline contextual bandits.

**W5. Missing references.** The paper omits directly relevant work on pessimistic offline contextual bandits, including [1, 2, 3] and many more. Authors should include these papers and the references therein. To find such papers, authors can read [3], which provides a unified framework citing many pessimistic approaches. While these works consider policy classes and are thus parametric with respect to the policy (though nonparametric with respect to rewards, except [2]), they remain relevant.

[1] https://arxiv.org/pdf/2006.10460 (AISTATS)

[2] https://www.researchgate.net/profile/Olivier-Jeunen/publication/353481957_Pessimistic_Reward_Models_for_Off-Policy_Learning_in_Recommendation/links/60ffe2ea1e95fe241a8ee976/Pessimistic-Reward-Models-for-Off-Policy-Learning-in-Recommendation.pdf (RecSys)


[3] https://arxiv.org/pdf/2406.03434 (UAI)

**W5. Missing discussion.** Numerous recent papers (e.g., [4, 5] and references therein) derive bounds that depend on coverage of $\pi^*$ rather than coverage of the whole action space. They are still different (e.g., parametric either on the reward or policy), and often assume the behavior policy is static (not adaptive). Discussing those would strengthen the position of the paper.

[4] https://arxiv.org/pdf/2402.14664 (AISTATS)

[5] https://arxiv.org/pdf/2309.15771 (ALT)

**Questions:**

**Q1.** Could the authors clarify the precise dependence of the theoretical bounds on the number of actions $K$?

**Q2.** What happens when the dataset already provides good coverage (large \$1/C^\star\$): will the algorithms become overly conservative in this regime?

**Q3.** Could the authors provide explicit analyses of the space complexity, computational cost, and inference time for both BIN-LCB and KNN-LCB?

**Q4.** Could the authors include experiments demonstrating cases where BIN-LCB or KNN-LCB outperform existing pessimistic offline learning methods?

**Q5.** A broader question: Is pessimism fundamentally necessary here? For example, some recent works [1, 2] suggest that greedy approaches outperform pessimistic ones in terms of average suboptimality (not worst-case suboptimality). Thus, using pessimism or not would depend on the setting (e.g., frequentist or Bayesian, etc.) and metric used (e.g., suboptimality or Bayesian suboptimality, etc.). Given that your algorithms incorporate pessimism, could you share your perspective on this?

[1] https://arxiv.org/pdf/2306.01237 (ICML)

[2] https://arxiv.org/pdf/2402.14664 (AISTATS)

---

> ### Author Response · Authors · 2025-11-20
>
> Thank you for dedicating time to review our paper and for your detailed feedback.
>
> 1. “No empirical validation is provided,”
>
> We have added an empirical validation section to the revised paper; see Appendix A (Experiments).
>
> 2. “Limited practicality of the proposed algorithms.”
>
> We first present explicit analyses of the time and space complexity of our algorithms, followed by a discussion of practical strategies that further improve their efficiency.
>
> For BIN-LCB, the offline preprocessing requires a single pass through the dataset: each context is binned in $O(d)$ time and the corresponding arm statistics are updated in $O(1)$, yielding a total cost of $O(Nd)$. At evaluation time, identifying the bin of a new context takes $O(d)$ and computing pessimistic estimates for all $K$ arms takes $O(K)$. Thus, BIN-LCB’s per-query cost is $O(d + K)$, and the space complexity is $O(K M^d)$, where $M^d$ is the number of bins; under the theoretically optimal choice of $M$, this remains sublinear in $N$. Importantly, evaluation time does not grow with $N$.
>
> For KNN-LCB, evaluating a new context involves computing all distances in $O(Nd)$, sorting them in $O(N\log N)$, and computing cumulative reward and count statistics in $O(NK)$. The selection of $k(a)$ for each arm is also $O(NK)$. Therefore, the per-query complexity is $O(N(d + \log N + K))$, with space complexity $O(N(d + K))$.
>
> While the current presentation of KNN-LCB uses a full nearest-neighbor ordering with a naïve per-query cost linear in $N$, practical implementations are far more efficient. KNN-LCB only requires neighbors of a query point in increasing distance order, and this can be obtained efficiently by building a simple spatial index (e.g., a KD-tree or Ball tree) once on the dataset. Such indices typically reduce nearest-neighbor retrieval from linear time to something much closer to logarithmic or otherwise sublinear, leading to a per-query cost of $O(k_{\mathrm{max}}\log N )$, where $k_{\mathrm{max}} \ll N$ is a user-chosen neighborhood cutoff. The theory indicates that the optimal $k$ is roughly $N^{2\beta/(2\beta+d)}$, offering a reasonable guideline for setting $k_{\mathrm{max}}$. These choices reduce the effective computational load by several orders of magnitude without altering the algorithm’s behavior, ensuring that KNN-LCB is practical in real-world settings.
>
> 3. “Insufficient discussion of the coverage condition.”
>
> When the dataset already provides good coverage, pessimistic algorithms might become somewhat conservative. In this regime, the offline data contains a substantial fraction of optimal-arm pulls, so the empirical reward estimates may be less informative than the raw action counts. This makes it difficult to fully exploit the richness of the data without risking over-optimism. Developing minimax-tight guarantees under this case is challenging. A refined treatment of this regime is an interesting direction that we leave for future work.
>
> 4. “Unclear dependence on the number of actions.”
>
> For the upper bound, the dependence on $K$ enters only through the variable $c$ in the analysis. As shown in the proof of Theorem 3.1, $c$ scales as $c \asymp \sqrt{K}$; see the last line of Section 5 (Proof of main results). Substituting this into the final bound yields an overall dependence of order $K^{(1+\alpha)/2}$.
>
> For the lower bound, our construction is based on a two-armed nonparametric bandit problem. Hence, the lower bound does not introduce any additional $K$-dependence beyond the binary case.
>
> We also clarify that throughout the paper we work in the standard fixed-$K$ regime, i.e., $K = O(1)$, which is common in the nonparametric contextual bandit literature. Under this assumption, the $K$ factor in the upper bound is absorbed into constants.
>
> 5. “Missing references.”
>
> Thanks for bringing these relevant work to our attention. We will add them to the related work in the revision.
>
> 6. “Missing discussion.”
>
> Thank you for the great suggestion. We will include a brief discussion of these connections in the revision to clarify the relationship and strengthen the positioning.
>
> 7. “A broader question: Is pessimism fundamentally necessary here?”
>
> We appreciate the reviewer’s question about the necessity of pessimism. In our setting, which focuses on frequentist minimax suboptimality under relaxed coverage, pessimism plays a crucial role. It is the standard and, to our knowledge, necessary tool for controlling extrapolation error in poorly supported regions and for achieving minimax-optimal guarantees, as reflected by the effective sample size $N / C^\star$.
>
> The cited works (Petrik et al., ICML’24; Aouali et al., AISTATS’25) show that pessimism can be suboptimal for Bayesian or average-case metrics when informative priors are available. Our results are complementary: in those regimes, greedy or posterior-mean methods can indeed be preferable, whereas for worst-case guarantees under partial coverage, pessimism remains the principled approach.

---

> > ### Comment · Reviewer_7jZZ · 2025-11-23
> >
> > I thank the authors for the rebuttal and the new **Appendix A**. The theoretical contribution remains solid, but I still have concerns about the empirical validation and the way practicality is presented.
> >
> > **1. Empirical Validation (Re: W1).**
> > The new experiments are still very limited and described only briefly. Standard practice in this area is to include: (1) controlled synthetic experiments to clearly show when the method wins and when it fails; and (2) semi-synthetic benchmarks (e.g., supervised-to-bandit on UCI or recommender datasets) to test robustness in realistic high-dimensional regimes. Here, the evaluation is restricted to a synthetic setup with very minimal specification, and compares only to POEM and a greedy variant of the proposed method. There is still no empirical comparison to recent methods.
> >
> > **2. Complexity and Practicality (Re: W2).**
> > I disagree with considering the methods practical; the main computational limitations should be explicitly stated as **limitations** in the main text:
> >
> > * **BIN-LCB:** Space complexity is $O(K M^d)$, which strongly restricts applicability beyond low-dimensional settings.
> >
> > * **KNN-LCB:** The rebuttal argues that spatial indices (KD-trees, Ball trees) “typically” yield sublinear queries. Unless I am mistaken, classical results [1] show that in moderate to high dimensions, such indices often degenerate to near-linear scan, so sublinear behavior cannot be assumed outside low-dimensional regimes. The authors should clearly state that efficiency is limited to low-$d$ and $N$, rather than suggesting general real-world practicality.
> >
> > There is an extra page available for the final version. I suggest using it to clearly state these limitations and to integrate the additional discussions and clarifications from the rebuttal on coverage, $K$-dependence, pessimism, and related work.
> >
> > In summary, I believe the practicality is limited and should be discussed much more clearly. For this reason, I will keep my score, although I would be open to acceptance if the reviewer discussion reaches a different conclusion.
> >
> > [1] Weber, Roger, Hans-Jörg Schek, and Stephen Blott. "A quantitative analysis and performance study for similarity-search methods in high-dimensional spaces." VLDB. Vol. 98. 1998.

---

### Meta-Review · Area_Chair_tmDF · 2026-01-12

**Summary:**

This paper develops a nonparametric theoretical framework for offline policy learning in contextual bandits under relaxed coverage.

The reviewers have concerns regarding (i) no empirical evaluations, (ii) high time/space complexity of the algorithms, (iii) incomplete discussion on the coverage parameter, and (iv) assumption of known smoothness parameter.

**Reviewer Concerns:**

The authors provided responses which provided discussions on the coverage parameter and assumption of known smoothness parameter, and conducted a preliminary empirical evaluation. However, as observed by the reviewers, the empirical evaluations are very limited and described only briefly, the high time/space complexity of the algorithms still seem to be a valid concern. Given the high bar of ICLR, I would suggest rejection.

**Reviewer Scores:**

I believe all reviewers will keep their scores.

---

### Decision · Program_Chairs · 2026-01-26

Reject